# *ALPS*: Improved Optimization for Highly Sparse One-Shot Pruning for Large Language Models

**Xiang Meng**
Operations Research Center
Massachusetts Institute of Technology
mengx@mit.edu

**Kayhan Behdin**
Operations Research Center
Massachusetts Institute of Technology
behdink@mit.edu

**Haoyue Wang**
Operations Research Center
Massachusetts Institute of Technology
haoyuew@mit.edu

**Rahul Mazumder**
Operations Research Center
Massachusetts Institute of Technology
rahulmaz@mit.edu

## Abstract

The impressive performance of Large Language Models (LLMs) across various natural language processing tasks comes at the cost of vast computational resources and storage requirements. One-shot pruning techniques offer a way to alleviate these burdens by removing redundant weights without the need for retraining. Yet, the massive scale of LLMs often forces current pruning approaches to rely on heuristics instead of optimization-based techniques, potentially resulting in suboptimal compression. In this paper, we introduce *ALPS*, an optimization-based framework that tackles the pruning problem using the operator splitting technique and a preconditioned conjugate gradient-based post-processing step. Our approach incorporates novel techniques to accelerate and theoretically guarantee convergence while leveraging vectorization and GPU parallelism for efficiency. *ALPS* outperforms state-of-the-art methods in terms of the pruning objective and perplexity reduction, particularly for highly sparse models. On the LLaMA3-8B model with 70% sparsity, *ALPS* achieves a 29% reduction in test perplexity on the WikiText dataset and a 8% improvement in zero-shot benchmark performance compared to existing methods. Our code is available at `https://github.com/mazumder-lab/ALPS`.

## 1 Introduction

Large Language Models (LLMs) have revolutionized the field of natural language processing, demonstrating remarkable performance across a wide spectrum of tasks, from question answering and text generation to sentiment analysis and named entity recognition [Wei et al., 2022, Bubeck et al., 2023, Achiam et al., 2023]. The success of LLMs can in part be attributed to their massive scale—state-of-the-art models like OPT-175B [Zhang et al., 2022a] and LLaMA3 [Dubey et al., 2024] have hundreds of billions of parameters. However, this enormous size comes at a steep cost in terms of storage and computational resources. For instance, the OPT-175B model requires at least 320 GB of memory to store its parameters in half-precision (FP16) format, necessitating the use of multiple high-end GPUs for inference [Frantar and Alistarh, 2023]. To make LLMs more accessible and efficient, considerable efforts have been made to compress these models, with a particular emphasis on model quantization techniques [Lin et al., 2023, Behdin et al., 2023, Dettmers et al., 2023].

Network pruning [LeCun et al., 1989, Hassibi and Stork, 1992, Han et al., 2015], a complementary approach to quantization, has received comparatively less attention in the realm of LLMs. Pruning

aims to reduce the model size by identifying and removing redundant or less important weights, resulting in a sparser and more efficient network. Traditional pruning methods rely on iterative retraining to recover accuracy after each pruning stage [Han et al., 2015, Luo et al., 2017, Molchanov et al., 2016, Liu et al., 2018], which can be computationally expensive and time-consuming. To address this, recent research has focused on `one-shot` pruning methods [He et al., 2017, Singh and Alistarh, 2020] that compress a pre-trained model using only a small amount of data (e.g., a few thousand samples)—the key idea here is to perform pruning while retaining model accuracy as much as possible without expensive fine-tuning/retraining on the entire dataset. Many prior works [Frantar and Alistarh, 2022, Yu et al., 2022, Benbaki et al., 2023] on one-shot pruning address such pruning-accuracy tradeoffs using optimization based approaches.

Despite the progress made in one-shot pruning, the massive scale of LLMs poses additional challenges, as many one-shot pruning methods designed for vision models cannot be directly applied due to their large model sizes. To overcome this, existing LLM pruning methods often rely on heuristic approaches to prune instead of solving optimization problems. For instance, SparseGPT [Frantar and Alistarh, 2023] approximates the OBS [Hassibi and Stork, 1992] algorithm by employing partial weight updates and adaptive mask selection to reduce costly Hessian computation. Similarly, Wanda [Sun et al., 2023] prunes weights based on the product of their magnitudes and corresponding input activations. Zhang et al. [2023] propose to iteratively grow and prune the weight mask according to the change in reconstruction error achieved by each update. While these heuristics enable pruning at scale, they may lead to suboptimal compression (and hence, suboptimal compression-accuracy tradeoffs) compared to advanced optimization-based approaches, as we show in this paper.

In this paper, we propose *ALPS*[1], an optimization-based framework for one-shot LLM pruning. *ALPS* consists of two key components. First, it formulates pruning LLMs as an $\ell_0$-constrained optimization problem and solves it directly using the operator splitting technique (i.e., ADMM) [Boyd et al., 2011, Davis and Yin, 2016] without any simplification. The proposed algorithm simultaneously finds the support[2] of the weights and updates them. After the support stabilizes, *ALPS* fixes the support and employs preconditioned conjugate gradient (PCG) [Nocedal and Wright, 1999, Section 5] to compute the optimal weights on the support. Our modified PCG leverages sparse matrix structure (arising from pruning) and GPU computation to solve large systems efficiently, providing a significant speed advantage over direct matrix inversion. An outline of our proposed optimization-based framework *ALPS* is given in Figure 1. Compared to previous heuristics, *ALPS* offers higher quality supports and weights, as demonstrated in Section 4.1. This improvement translates to a better performance of the pruned model compared to existing methods, particularly in the challenging high-sparsity regime.

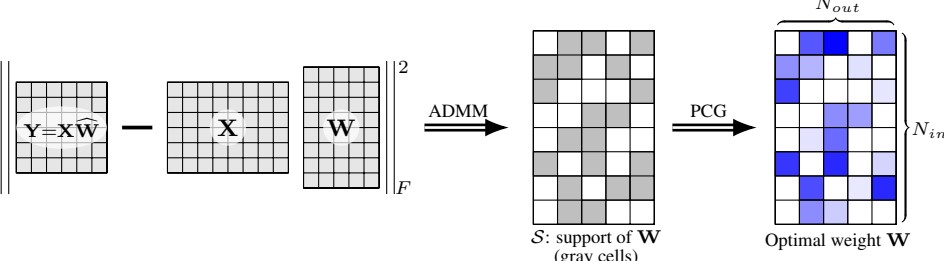

Figure 1: Overview of the proposed *ALPS* algorithm. (**Left**) The pruning problem with a layerwise reconstruction objective and an $\ell_0$ constraint on the weights (Section 3.1). (**Middle**) ADMM with a $\rho$-update scheme (Algorithm 1) is employed to determine high-quality support for the weight matrix **W** (Section 3.2). (**Right**) The optimization problem is restricted to the obtained support, and a modified PCG method (Algorithm 2) is used to solve for the optimal weight values within the support (Section 3.3).

**Contributions.** Our technical contributions are:

1. We introduce *ALPS*, a novel one-shot LLM pruning framework that formulates an $\ell_0$-constrained optimization problem with a layer-wise reconstruction objective. By extending the operator splitting technique (i.e., ADMM) to this non-convex, non-continuous problem, *ALPS* simultaneously

---

[1]**ADMM**-based **LLM P**runing in one-**S**hot

[2]The "support" of the weights refers to the set of indices corresponding to non-zero weights within a layer.

finds a high-quality support and updates the weights on the support. This approach leads to improvements over state-of-the-art heuristics in terms of the pruning objective. Furthermore, we provide theoretical convergence guarantees for our proposed algorithm, which, to the best of our knowledge, is a novel convergence result for $\ell_0$-constrained problems.

2. We further enhance the performance of *ALPS* through two techniques. First, we design a novel penalty parameter updating scheme that enables *ALPS* to find better support and accelerates its convergence. Second, we propose a post-processing technique to further improve the performance of the pruned models—we fix the support determined by ADMM and optimally solve the resulting quadratic problem using the PCG method. We utilize vectorization to solve the problem in a single pass and leverage GPU parallelism to further accelerate PCG. Our proposed method achieves a 20x-200x speedup compared to the vanilla backsolve approach.

3. *ALPS* substantially improves upon state-of-the-art methods for one-shot unstructured pruning of LLMs. For the LLaMA3-8B model with 70% sparsity, *ALPS* achieves a 29% reduction in test perplexity on the WikiText dataset and a 4%-13% improvement in performance on zero-shot benchmark evaluations. We also adapt *ALPS* to the popular N:M sparsity format [Zhou et al., 2021] and observe a 3%-10% higher performance compared to existing methods. Our code is publicly available at: `https://github.com/mazumder-lab/ALPS`.

## 2 Related Work

**Network pruning.** Network pruning is a well-established technique for reducing the complexity of deep neural networks by removing redundant weights [LeCun et al., 1989, Han et al., 2015]. Pruning methods can be classified based on the structure of the resulting sparse network and the training requirements. In terms of structure, pruning can be categorized into unstructured pruning, which removes individual weights [Han et al., 2015, Guo et al., 2016], and structured pruning, which removes entire structures such as channels, filters, or attention heads [Lebedev and Lempitsky, 2016, Wen et al., 2016, Voita et al., 2019, El Halabi et al., 2022]. Unstructured pruning offers better flexibility and higher sparsity levels but requires specialized hardware for acceleration, while structured pruning is more hardware-friendly but may suffer from larger performance loss. Based on the training requirements, pruning methods can be classified into three categories: (i) one-shot pruning, which directly removes weights from a pre-trained model without further training [Gale et al., 2019, Frantar and Alistarh, 2022, Meng et al., 2024a,b], (ii) gradual pruning, which begins with a pre-trained model but alternates between pruning and fine-tuning via SGD to recover performance [Molchanov et al., 2016, Zhu and Gupta, 2017, Blalock et al., 2020, Kurtic et al., 2022], and (iii) training from scratch, where the model is trained from randomly initialized weights, and the sparse network structure is either determined before training or evolves during the training process, [Mocanu et al., 2018, Dettmers and Zettlemoyer, 2019, Evci et al., 2020, Kusupati et al., 2020, Chen et al., 2021]. In this paper, we focus on one-shot unstructured pruning.

**Post-training unstructured pruning.** Based on their pruning objectives, there are three types of post-training unstructured pruning methods: (i) Importance-based methods, which assign a score to each weight (e.g., its absolute value) to assess its significance and decide whether it should be eliminated [Han et al., 2015, Lee et al., 2018, Molchanov et al., 2019, Sun et al., 2023]. (ii) Second-order techniques, which consider a local quadratic approximation of the loss function around the pre-trained model and remove weights based on their influence on the loss [Hassibi and Stork, 1992, Singh and Alistarh, 2020, Yu et al., 2022, Benbaki et al., 2023]. These approaches employ the empirical Fisher information matrix to estimate the Hessian matrix efficiently. (iii) Layer-wise pruning algorithms, which adapt OBS [Hassibi and Stork, 1992] framework to the layer-wise reconstruction objective [Dong et al., 2017, Frantar and Alistarh, 2022, 2023]. These methods prune each layer separately to address the computational challenge of calculating the full Hessian required in OBS. This work considers layer-wise reconstruction error as the pruning objective.

**Unstructured pruning in LLMs.** While pruning algorithms designed for convolutional networks [Singh and Alistarh, 2020, Chen et al., 2020, Frantar and Alistarh, 2022] can be readily adapted to moderate-sized language models like BERT [Vaswani et al., 2017], pruning LLMs with billions of parameters presents distinct challenges. The immense model size and extensive datasets associated with LLMs render traditional pruning methods computationally infeasible [Ma et al., 2023]. SparseGPT [Frantar and Alistarh, 2023] utilizes partial weight updates and adaptive mask selection to mitigate the expensive Hessian computation, while Wanda [Sun et al., 2023] directly obtains a

sparse LLM model using a criterion that considers the product of the absolute values of weights and their activations. DSnoT [Zhang et al., 2023] iteratively grow and prune the weight mask according to the change in reconstruction error achieved by each update. [Boža, 2024] introduces an efficient approach to determine the optimal weights on a given support and extends this technique to develop heuristics for updating the support.

**ADMM in network pruning.** The operator-splitting technique [Boyd et al., 2011, Davis and Yin, 2016] (also known as Alternating Direction Method of Multipliers, ADMM) is a well-known approach for solving composite optimization or optimization problems with coupled variables (or constraints), and has been used earlier in network pruning. Ye et al. [2018] applied ADMM to solve the original loss function under sparsity constraint, and Boža [2024] used ADMM to solve a convex pruning problem with fixed support. Moreover, Zhang et al. [2018] employed ADMM to train deep neural networks under sparsity constraints, while Ye et al. [2019] utilized ADMM to perform concurrent adversarial training and weight pruning. Our proposed method differs significantly from previous methods in two key aspects: (i) *ALPS* solves the pruning problem with an $\ell_0$ constraint at LLM scale, simultaneously optimizes over the weights and the sparsity pattern. (ii) We introduce a novel penalty parameter update scheme that ensures convergence both practically and theoretically.

# 3  *ALPS*: Effective LLM pruning in One-shot

## 3.1  Problem formulation

A common approach in post-training unstructured pruning of LLMs is to decompose the full-model compression problem into layer-wise subproblems. The quality of the solution for each subproblem is assessed by measuring the $\ell_2$ error between the output of the dense layer and that of the pruned one, given a set of input activations.

Formally, let $\widehat{\mathbf{W}} \in \mathbb{R}^{N_{in} \times N_{out}}$ denote the (dense) weight matrix of layer $\ell$, where $N_{in}$ and $N_{out}$ denote the input and output dimension of the layer, respectively. Given a set of $N$ calibration samples, the input activations can be represented as $\mathbf{X} \in \mathbb{R}^{NL \times N_{in}}$, where $L$ is the sequence length. The goal of pruning is to find a sparse weight matrix $\mathbf{W}$ that minimizes the reconstruction error between the original and pruned layer outputs, while satisfying a target sparsity constraint. In addition, we add a ridge term that penalizes the distance between $\mathbf{W}$ and $\widehat{\mathbf{W}}$, preventing $\mathbf{W}$ from diverging too far from the original weights. This layer-wise pruning problem can be formulated as an $\ell_0$-constrained optimization problem:

$$\min_{\mathbf{W} \in \mathbb{R}^{N_{in} \times N_{out}}} \ \|\mathbf{X}\widehat{\mathbf{W}} - \mathbf{X}\mathbf{W}\|_F^2 + \lambda_2\|\widehat{\mathbf{W}} - \mathbf{W}\|_F^2 \quad \text{s.t.} \quad \|\mathbf{W}\|_0 \le k, \tag{1}$$

where $\lambda_2 \ge 0$ and $\|\cdot\|_0$ denotes the $\ell_0$-(pseudo)norm, which counts the number of non-zero elements.

## 3.2  Operator-splitting for layer-wise pruning

Optimization of Problem (1) is quite challenging: we need to simultaneously find a support of $\mathbf{W}$ and a corresponding set of optimal weights (that minimize the objective restricted to the support). Notably, $\mathbf{W}$ may contain over 100 million parameters in the LLM setting, making (1) even more computationally demanding. To address this, we employ an operator-splitting technique [Boyd et al., 2011, Davis and Yin, 2016] (also known as ADMM), which decomposes the problem into two computationally 'friendlier' subproblems. Specifically, we reformulate problem (1) by introducing a copy $\mathbf{D}$ of weight matrix $\mathbf{W}$:

$$\min_{\mathbf{W}, \mathbf{D} \in \mathbb{R}^{N_{in} \times N_{out}}} \ \|\mathbf{X}\widehat{\mathbf{W}} - \mathbf{X}\mathbf{W}\|_F^2 + \lambda_2\|\widehat{\mathbf{W}} - \mathbf{W}\|_F^2 + \infty \cdot \mathbf{1}_{\|\mathbf{D}\|_0 > k} \quad \text{s.t.} \quad \mathbf{W} = \mathbf{D}, \tag{2}$$

where the penalty function "$\infty \cdot \mathbf{1}_{\|\mathbf{D}\|_0 > k}$" imposes the $\ell_0$ constraint $\|\mathbf{D}\|_0 \le k$ by assigning a value of zero when this condition is met and infinity otherwise. This reformulation separates the objective function into two independent parts while coupling the variables $\mathbf{W}$ and $\mathbf{D}$ through the linear constraint $\mathbf{W} = \mathbf{D}$. We consider the augmented Lagrangian function of this problem:

$$L_\rho(\mathbf{W}, \mathbf{D}, \mathbf{V}) = \|\mathbf{X}\widehat{\mathbf{W}} - \mathbf{X}\mathbf{W}\|_F^2 + \lambda_2\|\widehat{\mathbf{W}} - \mathbf{W}\|_F^2 + \infty \cdot \mathbf{1}_{\|\mathbf{D}\|_0 > k} + \langle \mathbf{V}, \mathbf{W} - \mathbf{D} \rangle + \frac{\rho}{2}\|\mathbf{W} - \mathbf{D}\|_F^2,$$
$$\tag{3}$$

where $\rho > 0$ is the quadratic penalty parameter. We minimize the augmented Lagrangian with respect to $\mathbf{W}$ and $\mathbf{D}$ alternatively, followed by a dual update. We get the following update at iteration $t$:

$$
\begin{aligned}
\mathbf{W}^{(t+1)} &= \arg\min_{\mathbf{W}} L_\rho(\mathbf{W}, \mathbf{D}^{(t)}, \mathbf{V}^{(t)}) = (\mathbf{H} + \rho\mathbf{I})^{-1}\left(\mathbf{H}\widehat{\mathbf{W}} - \mathbf{V}^{(t)} + \rho\mathbf{D}^{(t)}\right), \\
\mathbf{D}^{(t+1)} &= \arg\min_{\mathbf{D}} L_\rho(\mathbf{W}^{(t+1)}, \mathbf{D}, \mathbf{V}^{(t)}) = P_k\left(\mathbf{W}^{(t+1)} + \mathbf{V}^{(t)}/\rho\right), \\
\mathbf{V}^{(t+1)} &= \mathbf{V}^{(t)} + \rho(\mathbf{W}^{(t+1)} - \mathbf{D}^{(t+1)}),
\end{aligned}
\tag{4}
$$

where $\mathbf{H} = \mathbf{X}^\top\mathbf{X} + \lambda_2\mathbf{I}$. Here, the $\mathbf{W}$-update aims to minimize the objective by solving a system of equations, while the $\mathbf{D}$ update enforces sparsity by using the projection operator $P_k(\cdot)$, which projects an input matrix onto the set of matrices with at most $k$ non-zero elements. The dual update on matrix $\mathbf{V}$ ensures consistency between $\mathbf{W}$ and $\mathbf{D}$. As the iterations (4) progress, our proposed method concurrently identifies the support of the weight matrix and updates the weights on the determined support.

**$\rho$ update scheme.** In practice, we observe that the sequence of updates (4) and the resulting solution can be sensitive to the choice of the penalty parameter $\rho$. A small $\rho$ leads to slow convergence due to large changes in the support of $\mathbf{D}$ across iterations, while a large $\rho$ may compromise solution quality though the support stabilizes early on. To balance support quality and convergence speed, we introduce a novel penalty parameter update scheme. Starting with a small $\rho$, we gradually increase it every few iterations, with the increase rate proportional to the change in the support of $\mathbf{D}$. The detailed $\rho$ update scheme is provided in Appendix B.1. This scheme allows our algorithm to explore and find a good support when $\rho$ is small and to converge rapidly as $\rho$ grows, as demonstrated experimentally in Appendix B.2.1.

Algorithm 1 outlines the proposed operator-splitting technique with the $\rho$ update scheme. The convergence of Algorithm 1 is guaranteed by the following theorem, with its proof provided in Appendix A. We note that existing convergence results for operator-splitting type methods (e.g., ADMM) focus on convex or continuous problems [Hong et al., 2016, Wang et al., 2019]. However, our result guarantees the convergence on a non-convex, non-continuous $\ell_0$-constrained problem, which, to the best of our knowledge, is a novel convergence result for such a problem.

**Theorem 1.** *Let $\left\{\mathbf{D}^{(t)}\right\}_{t=0}^\infty$ and $\left\{\mathbf{W}^{(t)}\right\}_{t=0}^\infty$ be the sequences generated in Algorithm 1. Suppose the penalty parameter $\{\rho_t\}_{t=1}^\infty$ chosen in Algorithm 1 satisfies $\sum_{t=1}^\infty 1/\rho_t < \infty$. It then holds*

$$
\max\left\{\|\mathbf{D}^{(t+1)} - \mathbf{D}^{(t)}\|_F, \|\mathbf{W}^{(t+1)} - \mathbf{D}^{(t+1)}\|_F\right\} \leq C/\rho_t,
\tag{5}
$$

*where $C$ is a constant depending on $\mathbf{X}$, $\widehat{\mathbf{W}}$, $\lambda_2$, and $\sum_{t=1}^\infty 1/\rho_t$. In particular, there exists a matrix $\bar{\mathbf{D}}$ such that $\mathbf{D}^{(t)} \to \bar{\mathbf{D}}$ and $\mathbf{W}^{(t)} \to \bar{\mathbf{D}}$ as $t \to \infty$.*

---

**Algorithm 1** ADMM for layer-wise pruning with $\ell_0$ constraint

---

**Input:** Initial penalty $\rho_0$.
1: Initialize $\mathbf{V}^{(0)} = \mathbf{0}_{N_{in}\times N_{out}}$ and $\mathbf{D}^{(0)} = \mathbf{W}^{(0)} = \widehat{\mathbf{W}}$
2: **for** $t = 0, 1, \cdots$ **do**
3:     Update $\mathbf{W}^{(t+1)}, \mathbf{D}^{(t+1)}$ and $\mathbf{V}^{(t+1)}$ according to (4) with $\rho = \rho_t$.
4:     Increase $\rho_t$ to get $\rho_{t+1}$ based on the change in the support of $\text{Supp}\left(\mathbf{D}^{(t)}\right)$.
5: **end for**

---

**Computational cost.** The primary computational cost of Algorithm 1 arises from the $\mathbf{W}$-update step in display (4), which involves solving a system of linear equations. In the update, the inverse of $\mathbf{H} + \rho\mathbf{I}$ can be reused across iterations and needs to be updated when $\rho$ changes. To avoid re-computing the inverse, we store the eigenvalue decomposition $\mathbf{H} = \mathbf{Q}\mathbf{M}\mathbf{Q}^\top$. For varying $\rho$ values, the inverse can be efficiently calculated as $(\mathbf{H} + \rho\mathbf{I})^{-1} = \mathbf{Q}(\mathbf{M} + \rho\mathbf{I})^{-1}\mathbf{Q}^\top$, requiring only a single matrix-matrix multiplication. Additionally, the term $\mathbf{H}\widehat{\mathbf{W}}$ in $\mathbf{W}$-update remains constant across iterations and can be pre-computed and stored. Thus, each iteration of update (4) requires at most two matrix-matrix multiplications, leading to a time complexity of $O(N_{in}^2 N_{out})$.

**Extension to other sparsity patterns.** Algorithm 1 can be extended to support $N : M$ sparsity [Zhou et al., 2021, Hubara et al., 2021], a pattern in which a neural network has at most $N$ non-zero

weights in each group of $M$ consecutive weights. This sparsity pattern enables inference time acceleration on specialized hardware like NVIDIA A100 GPUs. To accommodate $N : M$ sparsity, we modify the $\mathbf{D}$-update step in (4) by replacing the projection operator $P_k(\cdot)$ with a projection onto the set of matrices satisfying $N : M$ sparsity. This modification can be easily implemented by applying magnitude pruning [Zhou et al., 2021] to $\mathbf{W}^{(t+1)} + \mathbf{V}^{(t)}/\rho$. Our approach can be further generalized to handle other structured sparsity patterns, such as block sparsity [Gray et al., 2017] and row sparsity [Meng et al., 2024b]. Similar to the $N : M$ sparsity adaptation, this is achieved by modifying the $\mathbf{D}$-update step to project onto the set of matrices satisfying the desired sparsity pattern. Importantly, our convergence results and the refining procedure introduced in Section 3.3 remain applicable to these various sparsity patterns.

## 3.3 Efficiently refining weights after support stabilization

Our proposed $\rho$-update technique enables Algorithm 1 to search for a high-quality support when $\rho$ is small, and the support stabilizes quickly as $\rho$ increases. However, once the support stabilizes, the convergence rate of Algorithm 1 becomes slow in practice. To accelerate the optimization process on the support, we employ a Preconditioned Conjugate Gradient (PCG) [Nocedal and Wright, 1999, Section 5] method with GPU parallelism and vectorization for efficient computation.

Formally, we introduce a post-processing technique that fixes the support $\mathcal{S}$ of the current solution $\mathbf{W}$ and refines the solution within this support, leading to the following problem:

$$\min_{\mathbf{W} \in \mathbb{R}^{N_{in} \times N_{out}}} \|\mathbf{X}\widehat{\mathbf{W}} - \mathbf{X}\mathbf{W}\|_F^2 + \lambda_2 \|\widehat{\mathbf{W}} - \mathbf{W}\|_F^2 \quad \text{s.t. } \mathrm{Supp}(\mathbf{W}) \subset \mathcal{S}. \tag{6}$$

(6) decomposes into separate least squares problems across the columns of $\mathbf{W}$. However, as illustrated in Figure 1 (Middle), the supports of the columns of $\mathbf{W}$ are different. Using direct matrix inversion (backsolve) to solve these problems would involve solving $N_{out}$ linear equations, each requiring the inversion of a submatrix of $\mathbf{H} = \mathbf{X}^\top \mathbf{X} + \lambda_2 \mathbf{I}$. Since the submatrices under consideration vary across different columns, parallelization is not straightforward, and we must solve $N_{out}$ different linear equations, each with size $O(N_{in})$. In LLMs, where $N_{out}$ and $N_{in}$ are of the order $10^4$, this would result in a significant computational expense. We present a workaround as discussed next.

To efficiently solve problem (6), we propose using the Preconditioned Conjugate Gradient (PCG) method, a high-performance numerical linear algebra technique for approximately solving systems of equations through repeated matrix-matrix multiplications. We further enhance PCG's performance by introducing two novel acceleration strategies. First, instead of solving for each column of $\mathbf{W}$ separately, we solve the entire problem in a single pass by directly solving the linear equation $\mathbf{H}\mathbf{W} = \mathbf{H}\widehat{\mathbf{W}}$ using PCG and projecting $\mathbf{W}$ onto the given support $\mathcal{S}$ in each iteration (Algorithm 2, line 8). We leverage vectorization to significantly enhance the speed. Second, we perform the matrix-matrix multiplications involved in PCG on the GPU, further utilizing GPU acceleration to expedite the computations. Algorithm 2 provides the detailed steps for the PCG method, and Figure1 offers an overview of our proposed *ALPS* method.

---

**Algorithm 2** PCG with vectorization for solving problem (6)

---

**Input:** Support $\mathcal{S}$, pre-conditioner $\mathbf{M} = \mathrm{Diag}(\mathbf{H})$, initial solution $\mathbf{W_0}$
  1: Set $\mathbf{R}_0 := \mathbf{H}(\widehat{\mathbf{W}} - \mathbf{W}_0)$
  2: Project $\mathbf{R}_0$ onto the support $\mathcal{S}$ by setting all elements outside the support to zero.
  3: Set $\mathbf{Z}_0 = \mathbf{M}^{-1}\mathbf{R}_0$ and $\mathbf{P}_0 = \mathbf{Z}_0$
  4: **for** $t = 0, 1, \ldots$ **do**
  5:     $\alpha_t = \mathrm{Tr}(\mathbf{R}_t^\top \mathbf{Z}_t)/\mathrm{Tr}(\mathbf{P}_t^\top \mathbf{H}\mathbf{P}_t)$
  6:     $\mathbf{W}_{t+1} = \mathbf{W}_t + \alpha_t \mathbf{P}_t$
  7:     $\mathbf{R}_{t+1} = \mathbf{R}_t - \alpha_t \mathbf{H}\mathbf{P}_t$
  8:     Project $\mathbf{R}_{t+1}$ onto the support $\mathcal{S}$ by setting all elements outside the support to zero.
  9:     $\mathbf{Z}_{t+1} = \mathbf{M}^{-1}\mathbf{R}_{t+1}$
 10:     **if** $\mathbf{R}_{t+1}$ is sufficiently small **then**
 11:         **break**
 12:     **end if**
 13:     $\beta_t = \mathrm{Tr}(\mathbf{R}_{t+1}^\top \mathbf{Z}_{t+1})/\mathrm{Tr}(\mathbf{R}_t^\top \mathbf{Z}_t)$
 14:     $\mathbf{P}_{t+1} := \mathbf{Z}_{t+1} + \beta_t \mathbf{P}_t$
 15: **end for**

---

# 4 Experimental Results

This section compares our proposed framework, *ALPS*, with state-of-the-art unstructured pruning methods for LLMs. Detailed information on the experimental setup and reproducibility is provided in Appendix B.1, while additional results are presented in Appendix B.2.

**Models and datasets.** We evaluate the performance of *ALPS* on the OPT model family [Zhang et al., 2022b] with sizes ranging from 1.3 billion to 30 billion parameters, the LLaMA2 model family [Touvron et al., 2023] with 7 billion and 13 billion parameters, and the LLaMA3 model [Dubey et al., 2024] with 8 billion parameters. Following the approach of Frantar and Alistarh, 2023, we use 128 segments of 2048 tokens each, randomly selected from the first shard of the C4 dataset [Raffel et al., 2020], as calibration data. We assess the performance using perplexity and zero-shot evaluation benchmarks, with perplexity calculated according to the procedure described by HuggingFace [Per, 2022], using full stride. The test sets of raw-WikiText2 [Merity et al., 2017], PTB [Marcus et al., 1994], and a subset of the C4 validation data, which are popular benchmarks in LLM pruning literature [Yao et al., 2022, Xiao et al., 2023, Meng et al., 2024b], are used for evaluation. Additionally, we consider five zero-shot tasks: MMLU [Hendrycks et al., 2021], PIQA [Bisk et al., 2020], LAMBADA [Paperno et al., 2016], ARC-Easy and ARC-Challenge [Clark et al., 2018].

**Competing methods.** We compare *ALPS* with several one-shot pruning methods for LLMs, including (i) Magnitude Pruning (MP, [Han et al., 2015]), (ii) SparseGPT [Frantar and Alistarh, 2023], (iii) Wanda Sun et al. [2023], and (iv) DSnoT [Zhang et al., 2023].

## 4.1 Reconstruction error on a single layer

We first evaluate the performance of our proposed *ALPS* framework on a single layer. Specifically, we prune a linear layer in the OPT-13B model with input and output dimensions of $5120$ to various sparsity levels and compute the relative reconstruction error of the pruned weight $\mathbf{W}$ using $\|\mathbf{X}\widehat{\mathbf{W}} - \mathbf{X}\mathbf{W}\|_F^2/\|\mathbf{X}\widehat{\mathbf{W}}\|_F^2$. The results are shown in Figure 2. As demonstrated, *ALPS* achieves significantly lower reconstruction errors compared to other methods, especially at high sparsity levels. For instance, at a sparsity level of $0.8$, *ALPS* yields a $7.6\%$ relative reconstruction error, while SparseGPT shows a $12\%$ error, and other methods exceed $20\%$. As demonstrated in Sections 4.2 and 4.3, our method's superior ability to approximate the dense model's output at each layer translates to much better performance in the pruned model.

We attribute the superior performance of *ALPS* in solving the reconstruction problem at each layer to two key aspects: (i) Algorithm 1 obtains a high-quality support by directly optimizing for an optimal subset of weights that contribute the most to recovering the dense model's output (ii) The PCG method in Algorithm 2 efficiently solves the reconstruction problem on a fixed support, further reducing the reconstruction error. To verify these claims, we conducted the following two ablation studies.

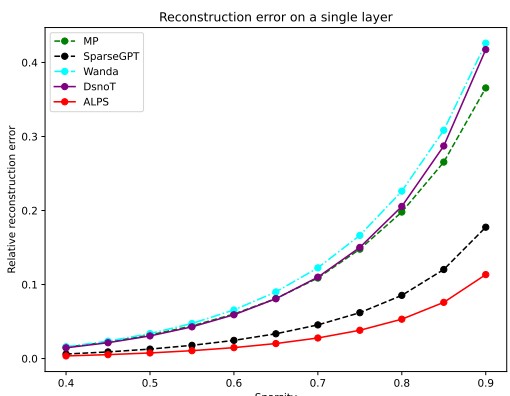

Figure 2: Performance analysis of pruning the "self_attn.k_proj" layer in the first block of the OPT-13B model at various sparsity levels. The plot shows the relative reconstruction error of pruned weights, comparing different pruning methods.

Firstly, we compare the quality of the support determined by various pruning methods. For each method, we prune the layer to different sparsity levels and fix the support of the weights matrix provided by the method. We then solve the post-processing problem (6) with this support to optimality and compute the relative reconstruction error of the resulting weights. This approach ensures that the reconstruction error depends solely on the quality of the support. Table 1 (left) presents the performance of each method. As shown, the support determined by *ALPS* yields $20\% \sim 40\%$ lower reconstruction error compared to other methods, demonstrating its effectiveness in finding high-quality supports.

| Sparsity | MP | SparseGPT | Wanda | DSnoT | *ALPS* |
|---|---|---|---|---|---|
| 0.5 | 1.10e-2 | 9.40e-3 | 1.17e-2 | 1.14e-2 | **7.56e-3** |
| 0.6 | 2.25e-2 | 1.88e-2 | 2.39e-2 | 2.38e-2 | **1.47e-2** |
| 0.7 | 4.38e-2 | 3.60e-2 | 4.65e-2 | 4.58e-2 | **2.78e-2** |
| 0.8 | 8.50e-2 | 6.95e-2 | 8.99e-2 | 9.02e-2 | **5.32e-2** |
| 0.9 | 1.78e-1 | 1.47e-1 | 1.87e-1 | 2.01e-1 | **1.13e-1** |

| Sparsity | w/o pp. | *ALPS* | | Backsolve | |
|---|---|---|---|---|---|
| | Error | Time(s) | Error | Time(s) | Error |
| 0.5 | 3.18e-2 | 0.77 | 1.24e-2 | 131 | 1.10e-2 |
| 0.6 | 6.01e-2 | 0.79 | 2.43e-2 | 95.0 | 2.25e-2 |
| 0.7 | 1.09e-1 | 0.78 | 4.56e-2 | 64.1 | 4.38e-2 |
| 0.8 | 1.98e-1 | 0.77 | 8.63e-2 | 37.8 | 8.50e-2 |
| 0.9 | 3.66e-1 | 0.76 | 1.78e-1 | 15.0 | 1.78e-1 |

Table 1: Performance analysis of pruning the "self_attn.k_proj" layer in the first block of the OPT-13B model at various sparsity levels. **(Left)** Relative reconstruction error $\|\mathbf{X}\widehat{\mathbf{W}} - \mathbf{X}\mathbf{W}\|_F^2 / \|\mathbf{X}\widehat{\mathbf{W}}\|_F^2$ of the optimal weights $\mathbf{W}$ constrained to the support determined by each pruning method. **(Right)** Comparison of time and reconstruction error for three scenarios, all using magnitude pruning to determine the support and then: (i) no post-processing (w/o pp.), (ii) refining the weights with *ALPS*, and (iii) refining the weights optimally with backsolve.

We then evaluate the effectiveness of our proposed post-processing method in finding optimal weights on a given support. We first apply magnitude pruning (MP) to determine the support of the weights and then consider three scenarios: (i) without post-processing (w/o pp.), (ii) using Algorithm 2 to solve (6) to refine the weights on the support (*ALPS*), and (iii) using PyTorch's `torch.linalg.solve` function to solve (6) to optimality (Backsolve). We assessed both the relative reconstruction error and the computational time for each scenario, with results presented in Table 1 (right). The findings demonstrate that the post-processing procedure significantly lowers the reconstruction error. Notably, our PCG method achieves errors comparable to the optimal solution but is 20x-200x faster, underscoring the efficiency and effectiveness of our approach.

## 4.2 Pruning OPT and LLaMA models

This section focuses on pruning OPT models and LLaMA models to various sparsity levels and evaluating the performance of the pruned models using perplexity and zero-shot benchmarks. The

| Model | Algorithm | WikiText2 ↓ | PTB ↓ | C4 ↓ | LAMBADA ↑ | PIQA ↑ | ARC-Easy ↑ | ARC-Challenge ↑ |
|---|---|---|---|---|---|---|---|---|
| OPT-1.3B | MP | 9409(±0) | 6689(±0) | 5652(±0) | 0.00(±0.00) | 52.12(±0.00) | 26.18(±0.00) | 20.90(±0.00) |
| | Wanda | 100.8(±3.8) | 128.4(±3.1) | 78.58(±2.34) | 9.77(±0.63) | 59.38(±0.46) | 37.92(±0.32) | 18.02(±0.36) |
| | SparseGPT | 52.02(±2.22) | 70.10(±3.31) | 37.05(±1.78) | 27.27(±1.04) | 62.38(±0.79) | 40.79(±0.58) | 19.69(±0.78) |
| | DSnoT | 367.5(±19.6) | 370.2(±30.6) | 205.4(±6.1) | 8.62(±0.23) | 56.90(±0.37) | 33.27(±0.47) | 17.49(±0.57) |
| | *ALPS* | **39.50**(±2.38) | **50.68**(±1.13) | **28.52**(±1.25) | **32.11**(±1.71) | **64.43**(±0.37) | **45.01**(±1.03) | **21.08**(±0.32) |
| OPT-2.7B | MP | 12249(±0) | 10993(±0) | 9960(±0) | 0.00(±0.00) | 52.94(±0.00) | 26.52(±0.00) | 19.71(±0.00) |
| | Wanda | 365.1(±16.5) | 379.4(±37.6) | 224.4(±4.8) | 5.02(±0.40) | 58.01(±0.38) | 34.85(±0.38) | 17.61(±0.28) |
| | SparseGPT | 28.93(±1.62) | 40.89(±1.19) | 23.11(±0.66) | 34.96(±1.97) | 66.50(±0.32) | 49.55(±0.50) | 21.67(±0.57) |
| | DSnoT | 114.8(±1.5) | 116.2(±6.7) | 75.28(±3.51) | 9.39(±0.63) | 59.62(±0.23) | 37.37(±0.48) | 18.43(±0.78) |
| | *ALPS* | **25.36**(±1.34) | **35.76**(±0.90) | **20.93**(±0.75) | **42.53**(±2.40) | **67.62**(±0.36) | **51.55**(±0.84) | **22.27**(±0.43) |
| OPT-6.7B | MP | 9970(±0) | 4779(±0) | 5055(±0) | 0.00(±0.00) | 52.67(±0.00) | 26.60(±0.00) | 21.16(±0.00) |
| | Wanda | 162.9(±8.7) | 204.9(±10.6) | 206.0(±13.5) | 2.82(±0.10) | 58.13(±0.16) | 35.82(±0.43) | 17.22(±0.39) |
| | SparseGPT | 21.14(±0.69) | 29.34(±0.44) | 19.07(±0.62) | 47.20(±0.94) | 69.23(±0.76) | 54.67(±0.20) | 24.08(±0.28) |
| | DSnoT | 7985(±465) | 6572(±783) | 4764(±478) | 0.15(±0.12) | 53.19(±0.36) | 29.13(±0.78) | 18.46(±0.48) |
| | *ALPS* | **18.99**(±0.85) | **24.89**(±0.26) | **17.01**(±0.60) | **56.04**(±1.32) | **71.39**(±0.27) | **58.52**(±0.74) | **26.19**(±0.52) |
| OPT-13B | MP | 524559(±0) | 146680(±0) | 155160(±0) | 0.00(±0.00) | 52.99(±0.00) | 25.25(±0.00) | 22.95(±0.00) |
| | Wanda | 63.42(±2.22) | 66.50(±1.93) | 50.19(±2.28) | 10.82(±0.95) | 63.03(±0.72) | 43.66(±0.52) | 23.50(±0.63) |
| | SparseGPT | 19.29(±0.42) | 25.50(±0.32) | 16.79(±0.45) | 47.97(±1.73) | 69.16(±0.31) | 53.51(±0.49) | 26.23(±0.96) |
| | DSnoT | 78.13(±2.94) | 60.11(±0.91) | 56.39(±1.79) | 14.43(±0.41) | 61.57(±0.60) | 40.52(±0.62) | 20.77(±0.67) |
| | *ALPS* | **16.71**(±0.62) | **21.49**(±0.40) | **15.07**(±0.46) | **57.08**(±1.62) | **71.73**(±0.22) | **59.30**(±0.50) | **27.97**(±0.93) |
| OPT-30B | MP | 26271(±0) | 12693(±0) | 13057(±0) | 0.00(±0.00) | 52.23(±0.00) | 25.46(±0.00) | 19.97(±0.00) |
| | Wanda | 10384(±198) | 5467(±104) | 6692(±191) | 0.01(±0.01) | 52.21(±0.27) | 25.61(±0.10) | 20.20(±0.41) |
| | SparseGPT | 13.61(±0.22) | 18.94(±0.25) | 13.96(±0.35) | 60.50(±0.88) | 73.74(±0.27) | 63.26(±0.44) | 28.81(±0.40) |
| | DSnoT | 11328(±368) | 5685(±248) | 6579(±298) | 0.04(±0.02) | 52.48(±0.30) | 26.36(±0.15) | 20.15(±0.31) |
| | *ALPS* | **12.67**(±0.23) | **17.47**(±0.27) | **13.00**(±0.32) | **63.52**(±1.38) | **75.05**(±0.38) | **65.52**(±0.75) | **30.55**(±0.76) |

Table 2: Performance analysis for one-shot unstructured pruning of OPT models (1.3B $\sim$ 30B) at 70% sparsity. We run each method five times and report the mean and standard deviation of each performance criterion. Here, ↓ denotes lower values corresponding to better performance, and ↑ denotes higher values corresponding to better performance.

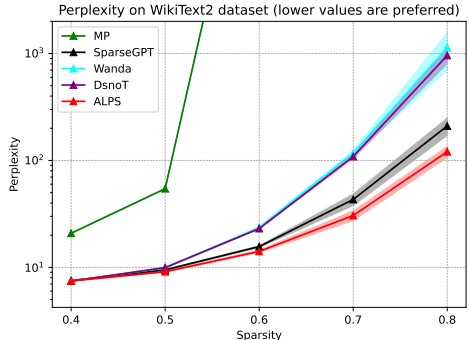 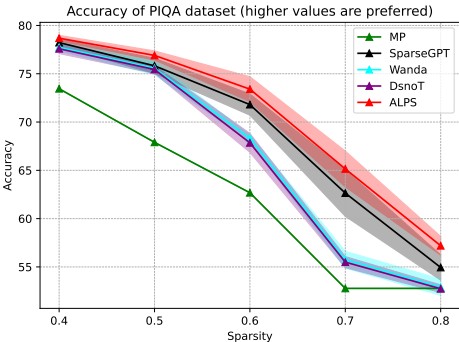

Figure 3: Performance analysis for one-shot unstructured pruning of LLaMA3-8B model at various sparsity levels on two datasets: WikiText2 **(Left)** and PIQA **(Right)**. We run each method five times and plot the shaded region as the area between the mean (solid line) and two standard deviations above and below the mean.

performance of the pruned LLaMA3-8B model at different sparsity levels on the WikiText2 and PIQA datasets is presented in Figure 3. Table 2 showcases the performance of OPT models with 70% sparsity on various datasets. Additional results on different models, sparsity levels, and datasets are provided in Appendix B.2.5.

Figure 3 demonstrates that *ALPS* outperforms other competitors when sparsity levels exceed 50%, and the performance gap between *ALPS* and other methods widens as the sparsity level increases. For instance, *ALPS* achieves a 60% perplexity reduction on the WikiText2 dataset compared to other methods at 80% sparsity level. This observation aligns with our findings in Section 4.1, confirming that *ALPS*'s highly advanced optimization method in solving layer-wise reconstruction problems enables it to better preserve performance at medium-to-high sparsity levels compared to other methods. Table 2 further validates this fact, showing that *ALPS* outperforms other methods by a large margin across all models on all criteria. This suggests the superiority of *ALPS* in pruning models at medium-to-high sparsity levels.

### 4.3 N:M sparsity

We further assess *ALPS*'s performance on $N : M$ sparsity patterns, with Table 3 listing the results for pruning OPT-30B and LLaMA2-13B models at 2:4 and 4:8 sparse patterns (see Appendix B.2.5 for other models). *ALPS* outperforms other methods on most datasets, achieving larger performance improvements in $N : M$ pruning compared to unstructured pruning at the same sparsity level. This is due to the higher complexity of the $N : M$ sparsity pruning problem, which *ALPS*, as a highly advanced optimization algorithm, can handle more effectively than competing heuristics.

## 5 Conclusion

We present *ALPS*, an efficient optimization-based framework for one-shot unstructured LLM pruning. *ALPS* employs the operator splitting technique to effectively solve the $\ell_0$-constrained layer-wise pruning problem. To enhance the performance of our algorithm, we introduce a novel penalty parameter updating scheme and a post-processing procedure using PCG with vectorization/GPU parallelism that takes into account problem-structure. We also establish novel convergence guarantees for our algorithm. *ALPS* can efficiently perform high-quality pruning of LLMs at scale. Our experiments confirm that *ALPS* outperforms existing pruning methods in terms of both the pruning objective and the performance of the pruned model. Future work will consider extending *ALPS* to incorporate structured pruning constraints and quantization to get a better understanding of the strengths and scope of our optimization-based approach.

| Model | Sparsity | Algorithm | WikiText2 ↓ | PTB ↓ | C4 ↓ | PIQA ↑ | ARC-Easy ↑ | ARC-Challenge ↑ |
|---|---|---|---|---|---|---|---|---|
| OPT-30B | 2:4 | MP | 1981(±0) | 2061(±0) | 1656(±0) | 58.22(±0.00) | 40.61(±0.00) | 18.94(±0.00) |
| | | Wanda | 13.23(±0.40) | 16.95(±0.28) | 14.67(±0.18) | 74.87(±0.14) | 64.21(±0.40) | 29.23(±0.37) |
| | | SparseGPT | 10.90(±0.07) | 14.02(±0.10) | 12.04(±0.14) | 75.59(±0.28) | 66.75(±0.55) | 31.23(±0.32) |
| | | DSnoT | 12.36(±0.10) | 15.73(±0.07) | 13.55(±0.08) | 74.94(±0.20) | 64.19(±0.19) | 29.44(±0.50) |
| | | ALPS | **10.64**(±0.09) | **13.75**(±0.08) | **11.69**(±0.15) | **75.93**(±0.09) | **66.82**(±0.54) | **31.45**(±0.37) |
| | 4:8 | MP | 564.1(±0.0) | 1487(±0) | 1005(±0) | 62.84(±0.00) | 42.47(±0.00) | 22.27(±0.00) |
| | | Wanda | 10.78(±0.08) | 14.07(±0.07) | 12.13(±0.03) | 75.65(±0.18) | 66.90(±0.46) | 30.48(±0.41) |
| | | SparseGPT | 10.30(±0.06) | 13.35(±0.15) | 11.52(±0.09) | 76.10(±0.25) | 67.88(±0.48) | **32.17**(±0.88) |
| | | DSnoT | 10.83(±0.08) | 13.93(±0.04) | 12.16(±0.03) | 75.42(±0.18) | 66.31(±0.44) | 30.58(±0.46) |
| | | ALPS | **10.14**(±0.05) | **13.18**(±0.12) | **11.32**(±0.09) | **76.61**(±0.50) | **67.89**(±0.25) | 32.01(±0.49) |
| LLaMA2-13B | 2:4 | MP | 8.89(±0.00) | 203.8(±0.0) | 10.80(±0.00) | 71.49(±0.00) | 57.66(±0.00) | 30.38(±0.00) |
| | | Wanda | 9.02(±0.04) | 88.93(±1.02) | 11.12(±0.05) | 73.16(±0.27) | 63.67(±0.54) | 33.92(±0.37) |
| | | SparseGPT | 8.76(±0.10) | 64.82(±3.16) | 10.06(±0.26) | 73.94(±0.60) | 65.10(±0.86) | 35.31(±0.57) |
| | | DSnoT | 9.21(±0.05) | 87.54(±2.04) | 11.43(±0.07) | 72.30(±0.23) | 63.38(±0.68) | 33.63(±0.63) |
| | | ALPS | **8.14**(±0.10) | **52.34**(±2.08) | **9.36**(±0.33) | **74.72**(±0.58) | **65.71**(±0.59) | **35.96**(±0.50) |
| | 4:8 | MP | 7.32(±0.00) | 137.3(±0.0) | 9.14(±0.00) | 74.43(±0.00) | 63.01(±0.00) | 35.67(±0.00) |
| | | Wanda | 7.02(±0.01) | 55.07(±0.79) | 8.84(±0.02) | 75.69(±0.31) | 67.05(±0.13) | 38.26(±0.56) |
| | | SparseGPT | 7.01(±0.03) | 47.64(±0.81) | 8.52(±0.14) | 75.84(±0.26) | 69.36(±0.84) | 38.74(±0.95) |
| | | DSnoT | 7.13(±0.02) | 54.07(±0.88) | 8.95(±0.02) | 75.64(±0.23) | 67.33(±0.42) | 37.58(±0.63) |
| | | ALPS | **6.81**(±0.07) | **42.15**(±0.89) | **8.22**(±0.17) | **76.52**(±0.57) | **69.41**(±0.69) | **39.22**(±0.56) |

Table 3: Performance analysis for one-shot pruning of OPT-30B and LLaMA2-13B at $2:4$ and $4:8$ sparsity patterns. We run each method five times and report the mean and standard deviation of each performance criterion. Here, ↓ denotes lower values correspond to better performance, and ↑ denotes higher values correspond to better performance.

## Acknowledgements

This research is supported in part by grants from ONR (N000142112841 and N000142212665). We acknowledge the MIT SuperCloud and Lincoln Laboratory Supercomputing Center for providing HPC resources that have contributed to the research results reported within this paper. Additionally, we thank Google for providing us with Google Cloud Credits. We thank Shibal Ibrahim, Mehdi Makni, and Gabriel Afriat for their helpful discussions.

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

# A Proofs of Theorem 1

***Proof.*** For the sake of conciseness, throughout the proof, we denote $\mathbf{H} = \mathbf{X}^\top\mathbf{X} + \lambda_2\mathbf{I}$ and $\mathbf{G} = \left(\mathbf{X}^\top\mathbf{X} + \lambda_2\mathbf{I}\right)\widehat{\mathbf{W}}$. To establish the theorem, we first present the following two lemmas. The proofs of these two lemmas are given in Section A.1 and A.2, respectively.

**Lemma 1.** *Let* $\left\{\mathbf{D}^{(t)}\right\}_{t=0}^{\infty}$ *and* $\left\{\mathbf{V}^{(t)}\right\}_{t=0}^{\infty}$ *be the sequence generated in Algorithm 1. Then for any* $t \geq 0$*, it holds*

$$\|\mathbf{V}^{(t+1)}\|_F \leq \|\mathbf{G} - \mathbf{H}\mathbf{D}^{(t)}\|_F + \frac{\|\mathbf{H}\mathbf{V}^{(t)}\|_F}{\rho_t} \tag{7}$$

*and*

$$\|\mathbf{D}^{(t+1)} - \mathbf{D}^{(t)}\|_F \leq \frac{2}{\rho_t}\left(\|\mathbf{G} - \mathbf{H}\mathbf{D}^{(t)}\|_F + \frac{\|\mathbf{H}\mathbf{V}^{(t)}\|_F}{\rho_t}\right). \tag{8}$$

**Lemma 2.** *Let* $\left\{\mathbf{D}^{(t)}\right\}_{t=0}^{\infty}$, $\left\{\mathbf{W}^{(t)}\right\}_{t=0}^{\infty}$ *and* $\left\{\mathbf{V}^{(t)}\right\}_{t=0}^{\infty}$ *be the sequence generated in Algorithm 1. Suppose* $\{\rho_t\}_{t=0}^{\infty}$ *is non-decreasing. Then for any* $t \geq 0$*, it holds*

$$\|\mathbf{D}^{(t)}\|_F + \frac{\|\mathbf{V}^{(t)}\|_F}{\rho_t} \leq \left[\prod_{s=0}^{t-1}\left(1 + \frac{3\|\mathbf{H}\|_2}{\rho_s}\right)\right] \cdot \left(\|\mathbf{D}^{(0)}\|_F + \frac{\|\mathbf{V}^{(0)}\|_F}{\rho_0} + \sum_{s=0}^{t-1}\frac{3\|\mathbf{G}\|_F}{\rho_s}\right) \tag{9}$$

Returning to the proof of the main theorem, combining Lemma 2 with the initialization of Algorithm 1 gives

$$
\begin{aligned}
\|\mathbf{D}^{(t)}\|_F + \frac{\|\mathbf{V}^{(t)}\|_F}{\rho_t} &\leq \left[\prod_{s=0}^{t-1}\left(1 + \frac{3\|\mathbf{H}\|_2}{\rho_s}\right)\right] \cdot \left(\|\mathbf{D}^{(0)}\|_F + \frac{\|\mathbf{V}^{(0)}\|_F}{\rho_0} + \sum_{s=0}^{t-1}\frac{3\|\mathbf{G}\|_F}{\rho_s}\right) \\
&\leq \exp\left(3\|\mathbf{H}\|_2\sum_{s=0}^{\infty}\frac{1}{\rho_s}\right) \cdot \left(\|\mathbf{G}\|_F + 3\|\mathbf{G}\|_F\sum_{s=0}^{\infty}\frac{1}{\rho_s}\right)
\end{aligned}
\tag{10}
$$

Let

$$C(\mathbf{X}, \widehat{\mathbf{W}}, \rho_0, t_u, \hat{\tau}) := 2\|\mathbf{G}\|_F + 2\|\mathbf{H}\|_2\left(\exp\left(3\|\mathbf{H}\|_2\sum_{s=0}^{\infty}\frac{1}{\rho_s}\right) \cdot \left(\|\mathbf{G}\|_F + 3\|\mathbf{G}\|_F\sum_{s=0}^{\infty}\frac{1}{\rho_s}\right)\right) \tag{11}$$

be the constant depending on $\mathbf{X}$, $\widehat{\mathbf{W}}$ and $\sum_{s=0}^{\infty}1/\rho_s$. Lemma 1 together with (10) leads to

$$
\begin{aligned}
\|\mathbf{V}^{(t+1)}\|_F &\leq \|\mathbf{G} - \mathbf{H}\mathbf{D}^{(t)}\|_F + \frac{\|\mathbf{H}\mathbf{V}^{(t)}\|_F}{\rho_t} \\
&\leq \|\mathbf{G}\|_F + \|\mathbf{H}\|_2\left(\|\mathbf{D}^{(t)}\|_F + \frac{\|\mathbf{V}^{(t)}\|_F}{\rho_t}\right) \leq \frac{1}{2}C(\mathbf{X}, \widehat{\mathbf{W}}, \rho_0, t_u, \hat{\tau})
\end{aligned}
\tag{12}
$$

and

$$\|\mathbf{D}^{(t+1)} - \mathbf{D}^{(t)}\|_F \leq \frac{2}{\rho_t}\left(\|\mathbf{G} - \mathbf{H}\mathbf{D}^{(t)}\|_F + \frac{\|\mathbf{H}\mathbf{V}^{(t)}\|_F}{\rho_t}\right) \leq \frac{C(\mathbf{X}, \widehat{\mathbf{W}}, \rho_0, t_u, \hat{\tau})}{\rho_t}. \tag{13}$$

It then follows from $\mathbf{W}^{(t+1)} - \mathbf{D}^{(t+1)} = (\mathbf{V}^{(t+1)} - \mathbf{V}^{(t)})/\rho_t$ that

$$\|\mathbf{W}^{(t+1)} - \mathbf{D}^{(t+1)}\|_F \leq \frac{\|\mathbf{V}^{(t+1)}\|_F + \|\mathbf{V}^{(t)}\|_F}{\rho_t} \leq \frac{C(\mathbf{X}, \widehat{\mathbf{W}}, \rho_0, t_u, \hat{\tau})}{\rho_t}. \tag{14}$$

Therefore, we prove the desired inequality. Since $\sum_{s=0}^{\infty}1/\rho_s < \infty$, $\{\mathbf{D}\}_{t=0}^{\infty}$ is a Cauchy sequence, and therefore there exists a matrix $\bar{\mathbf{D}}$ such that $\mathbf{D}^{(t)} \to \bar{\mathbf{D}}$. It follows from $\|\mathbf{W}^{(t+1)} - \mathbf{D}^{(t+1)}\|_F \to 0$ that $\mathbf{W}^{(t)} \to \bar{\mathbf{D}}$. The proof is completed.

$$\square$$

## A.1 Proof of Lemma 1

***Proof.*** According to the $\mathbf{W}-$update rule in (4), it holds

$$
\begin{aligned}
\mathbf{W}^{(t+1)} - \mathbf{D}^{(t)} + \frac{\mathbf{V}^{(t)}}{\rho^{(t)}} &= (\mathbf{H} + \rho_t\mathbf{I})^{-1}(\mathbf{G} - \mathbf{V}^{(t)} + \rho_t\mathbf{D}^{(t)}) - \mathbf{D}^{(t)} + \frac{\mathbf{V}^{(t)}}{\rho^{(t)}} \\
&= \left((\mathbf{H} + \rho_t\mathbf{I})^{-1}\rho_t - \mathbf{I}\right)\mathbf{D}^{(t)} + (\mathbf{H} + \rho_t\mathbf{I})^{-1}(\mathbf{G} - \mathbf{V}^{(t)}) + \frac{\mathbf{V}^{(t)}}{\rho_t} \\
&= -\frac{1}{\rho_t}\left(\mathbf{I} + \frac{\mathbf{H}}{\rho_t}\right)^{-1}\mathbf{H}\mathbf{D}^{(t)} + \frac{1}{\rho_t}\left(\mathbf{I} + \frac{\mathbf{H}}{\rho_t}\right)^{-1}(\mathbf{G} - \mathbf{V}^{(t)}) + \frac{\mathbf{V}^{(t)}}{\rho_t} \\
&= \frac{1}{\rho_t}\left(\mathbf{I} + \frac{\mathbf{H}}{\rho_t}\right)^{-1}(\mathbf{G} - \mathbf{H}\mathbf{D}^{(t)}) + \frac{1}{\rho_t}\left[\mathbf{I} - \left(\mathbf{I} + \frac{\mathbf{H}}{\rho_t}\right)^{-1}\right]\mathbf{V}^{(t)} \\
&= \frac{1}{\rho_t}\left(\mathbf{I} + \frac{\mathbf{H}}{\rho_t}\right)^{-1}\left(\mathbf{G} - \mathbf{H}\mathbf{D}^{(t)} + \frac{\mathbf{H}\mathbf{V}^{(t)}}{\rho_t}\right)
\end{aligned}
\tag{15}
$$

Therefore, we obtain

$$
\begin{aligned}
\left\|\mathbf{W}^{(t+1)} - \mathbf{D}^{(t)} + \frac{\mathbf{V}^{(t)}}{\rho^{(t)}}\right\|_F &\leq \frac{1}{\rho_t}\left\|\left(\mathbf{I} + \frac{\mathbf{H}}{\rho_t}\right)^{-1}\right\|_2\left\|\mathbf{G} - \mathbf{H}\mathbf{D}^{(t)} + \frac{\mathbf{H}\mathbf{V}^{(t)}}{\rho_t}\right\|_F \\
&\leq \frac{1}{\rho_t}\left\|\mathbf{G} - \mathbf{H}\mathbf{D}^{(t)} + \frac{\mathbf{H}\mathbf{V}^{(t)}}{\rho_t}\right\|_F \\
&\leq \frac{1}{\rho_t}\left(\|\mathbf{G} - \mathbf{H}\mathbf{D}^{(t)}\|_F + \frac{\|\mathbf{H}\mathbf{V}^{(t)}\|}{\rho_t}\right).
\end{aligned}
\tag{16}
$$

Denote $\widetilde{\mathcal{I}} := \{(i,j) \in [N_{in}] \times [N_{out}] \mid \mathbf{D}_{ij}^{(t)} = 0\}$. It follows from the $\mathbf{D}-$update rule and the definition of the projection operator that

$$
\begin{aligned}
\left\|\mathbf{D}^{(t+1)} - \mathbf{W}^{(t+1)} - \frac{\mathbf{V}^{(t)}}{\rho_t}\right\|_F^2 &= \min_{\substack{\mathcal{I} \subseteq [N_{in}] \times [N_{out}] \\ |\mathcal{I}| = N_{in}N_{out}-k}} \sum_{(i,j)\in\mathcal{I}} \left(\mathbf{W}^{(t+1)} + \frac{\mathbf{V}^{(t)}}{\rho_t}\right)_{i,j}^2 \\
&\leq \sum_{(i,j)\in\widetilde{\mathcal{I}}} \left(\mathbf{W}^{(t+1)} + \frac{\mathbf{V}^{(t)}}{\rho_t}\right)_{i,j}^2 = \sum_{(i,j)\in\widetilde{\mathcal{I}}} \left(\mathbf{W}^{(t+1)} - \mathbf{D}^{(t)} + \frac{\mathbf{V}^{(t)}}{\rho_t}\right)_{i,j}^2 \\
&\leq \left\|\mathbf{W}^{(t+1)} - \mathbf{D}^{(t)} + \frac{\mathbf{V}^{(t)}}{\rho_t}\right\|_F^2
\end{aligned}
\tag{17}
$$

Together with (16), we get

$$
\left\|\mathbf{D}^{(t+1)} - \mathbf{W}^{(t+1)} - \frac{\mathbf{V}^{(t)}}{\rho_t}\right\|_F \leq \frac{1}{\rho_t}\left(\|\mathbf{G} - \mathbf{H}\mathbf{D}^{(t)}\|_F + \frac{\|\mathbf{H}\mathbf{V}^{(t)}\|}{\rho_t}\right).
\tag{18}
$$

It then follows from the $\mathbf{V}-$update rule that

$$
\frac{\|\mathbf{V}^{(t+1)}\|_F}{\rho_t} = \left\|\mathbf{D}^{(t+1)} - \mathbf{W}^{(t+1)} - \frac{\mathbf{V}^{(t)}}{\rho_t}\right\|_F \leq \frac{1}{\rho_t}\left(\|\mathbf{G} - \mathbf{H}\mathbf{D}^{(t)}\|_F + \frac{\|\mathbf{H}\mathbf{V}^{(t)}\|}{\rho_t}\right)
\tag{19}
$$

This establishes the inequality (7). Furthermore, by summing up (16) and (18) and applying the triangle inequality, we verify the inequality (8). □

## A.2 Proof of Lemma 2

***Proof.*** It follows from Lemma 1 that

$$
\begin{aligned}
\|\mathbf{V}^{(t+1)}\|_F &\leq \|\mathbf{G} - \mathbf{H}\mathbf{D}^{(t)}\|_F + \frac{\|\mathbf{H}\mathbf{V}^{(t)}\|_F}{\rho_t} \\
&\leq \|\mathbf{H}\|_2\|\mathbf{D}^{(t)}\|_F + \|\mathbf{G}\|_F + \frac{\|\mathbf{H}\|_2\|\mathbf{V}^{(t)}\|_F}{\rho_t}
\end{aligned}
\tag{20}
$$

and

$$\|\mathbf{D}^{(t+1)} - \mathbf{D}^{(t)}\|_F \leq \frac{2}{\rho_t}\left(\|\mathbf{G} - \mathbf{H}\mathbf{D}^{(t)}\|_F + \frac{\|\mathbf{H}\mathbf{V}^{(t)}\|_F}{\rho_t}\right)$$
$$\leq \frac{2}{\rho_t}\left(\|\mathbf{H}\|_2\|\mathbf{D}^{(t)}\|_F + \|\mathbf{G}\|_F + \frac{\|\mathbf{H}\|_2\|\mathbf{V}^{(t)}\|_F}{\rho_t}\right). \tag{21}$$

This further implies

$$\|\mathbf{D}^{(t+1)}\|_F \leq \left(1 + \frac{2\|\mathbf{H}\|_2}{\rho_t}\right)\|\mathbf{D}^{(t)}\|_F + \frac{2\|\mathbf{G}\|_F}{\rho_t} + \frac{2\|\mathbf{H}\|_2\|\mathbf{V}^{(t)}\|_F}{\rho_t^2} \tag{22}$$

Combining inequalities (20) and (22) yields

$$\|\mathbf{D}^{(t+1)}\|_F + \frac{\|\mathbf{V}^{(t+1)}\|_F}{\rho_{t+1}} \leq \|\mathbf{D}^{(t+1)}\|_F + \frac{\|\mathbf{V}^{(t+1)}\|_F}{\rho_t}$$
$$\leq \left(1 + \frac{3\|\mathbf{H}\|_2}{\rho_t}\right)\|\mathbf{D}^{(t)}\|_F + \frac{3\|\mathbf{G}\|_F}{\rho_t} + \frac{3\|\mathbf{H}\|_2\|\mathbf{V}^{(t)}\|_F}{\rho_t^2} \tag{23}$$
$$\leq \left(1 + \frac{3\|\mathbf{H}\|_2}{\rho_t}\right)\left(\|\mathbf{D}^{(t)}\|_F + \frac{\|\mathbf{V}^{(t)}\|_F}{\rho_t}\right) + \frac{3\|\mathbf{G}\|_F}{\rho_t}$$

Denote $a_t := \|\mathbf{D}^{(t)}\|_F + \|\mathbf{V}^{(t)}\|_F/\rho_t$, then the above inequality can be rewritten as

$$a_{t+1} \leq \left(1 + \frac{3\|\mathbf{H}\|_2}{\rho_t}\right)a_t + \frac{3\|\mathbf{G}\|_F}{\rho_t} \tag{24}$$

Therefore,

$$\frac{a_{t+1}}{\prod_{s=0}^{t}(1 + 3\|\mathbf{H}\|_2/\rho_k)} \leq \frac{a_t}{\prod_{s=0}^{t-1}(1 + 3\|\mathbf{H}\|_2/\rho_k)} + \frac{3\|\mathbf{G}\|_F}{\rho_t \prod_{s=0}^{t}(1 + 3\|\mathbf{H}\|_2/\rho_k)}$$
$$\leq \frac{a_t}{\prod_{s=0}^{t-1}(1 + 3\|\mathbf{H}\|_2/\rho_k)} + \frac{3\|\mathbf{G}\|_F}{\rho_t} \tag{25}$$

It then follows from telescoping that

$$\frac{a_t}{\prod_{s=0}^{t-1}(1 + 3\|\mathbf{H}\|_2/\rho_k)} \leq a_0 + \sum_{s=0}^{t-1}\frac{3\|\mathbf{G}\|_F}{\rho_t} \tag{26}$$

Recalling the definition of $a_t$ completes the proof. $\qquad\square$

# B  Experimental Details

## B.1  Experimental setup

We performed all experiments on a computing cluster using an Intel Xeon Gold 6248 machine with 20 CPUs and a single NVIDIA V100 GPU, which is equipped with 192GB of CPU RAM and 32GB of CUDA memory. The PyTorch library Paszke et al. [2017] was used to implement all language models and pruning methods for our experiments.

**Pruning problem setup.** For a given sparsity $s$, we set the $\ell_0$ constraint $k$ in the pruning problem (1) to $\lfloor N_{in}N_{out}s \rfloor$. Following the pruning framework proposed by Frantar and Alistarh [2023], Meng et al. [2024b], we solve the LLM pruning problem sequentially, layer by layer. For layer $\ell$, the input activation matrix $\mathbf{X}$ in (1) is set as the output of the previous $\ell - 1$ pruned layers on $N$ calibration samples.

**Data pre-processing.** Let $\mathbf{E} = \text{Diag}(\mathbf{X}^\top\mathbf{X} + \lambda_2\mathbf{I})^{-1/2}$. To achieve better scaling, we define $\mathbf{W}' = \mathbf{E}^{-1}\mathbf{W}$ and reformulate Equation (1) into the following equivalent form:

$$\min_{\mathbf{W}'\in\mathbb{R}^{N_{in}\times N_{out}}} \|\mathbf{X}\widehat{\mathbf{W}} - \mathbf{X}\mathbf{E}\mathbf{W}'\|_F^2 + \lambda_2\|\widehat{\mathbf{W}} - \mathbf{E}\mathbf{W}'\|_F^2 \quad \text{s.t. } \|\mathbf{W}'\|_0 \leq k \tag{27}$$

Practically, we apply Algorithm 1 to solve Equation (27) and recover the solution to the original problem by setting $\mathbf{W} = \mathbf{E}\mathbf{W}'$. It is important to note that this pre-processing step does not alter

the procedure of Algorithm 1 or affect the convergence analysis. It only modifies the updates within Algorithm 1 and can lead to a better convergence rate in practice.

**Hyperparamter choice.** We choose $\lambda_2 = 0.01 \operatorname{Tr}(\mathbf{X}^\top \mathbf{X})$. In Algorithm 1, we set $\rho_0 = 0.1$. And we update $\rho$ every 3 iteration based on a step function that depends on the current value of $\rho_t$ and $s_t := |\operatorname{Supp}(\mathbf{D}^{(t)}) \Delta \operatorname{Supp}(\mathbf{D}^{(t-3)})|$, which represents the number of elements in the symmetric difference between $\operatorname{Supp}(\mathbf{D}^{(t)})$ and $\operatorname{Supp}(\mathbf{D}^{(t-3)})$. Specifically, we set

$$\rho_{t+1} = \begin{cases} 1.3\rho_t & \text{if } s_t \geq 0.1k, \\ 1.2\rho_t & \text{if } s_t \geq 0.005k, \\ 1.1\rho_t & \text{if } s_t \geq 1. \end{cases} \tag{28}$$

If $s_t = 0$, it indicates that $\rho$ is sufficiently large and the support has stabilized. In this case, we terminate Algorithm 1, set $\mathcal{S}$ as the support of $\mathbf{W}$, and apply Algorithm 2 with 10 iterations to solve problem (6) with support $\mathcal{S}$.

**Implementation details.** Below are the configuration and implementation details for the competing methods and our proposed framework *ALPS*.

- MP: For each layer in the LLM, we perform magnitude pruning by sorting the absolute values of all entries of the dense weight $\widehat{\mathbf{W}}$ in descending order, keeping the top $k$ entries unchanged, and setting the remaining entries to zero.
- SparseGPT: We utilize the authors' implementation (codes available on GitHub) with default hyperparameter settings to perform one-shot unstructured LLM pruning.
- Wanda: We utilize the authors' implementation (codes available on GitHub) with default hyperparameter settings to perform one-shot unstructured LLM pruning.
- DSnoT: We utilize the authors' implementation (codes available on GitHub) with default hyperparameter settings to perform one-shot unstructured LLM pruning.

## B.2 Additional experimental results

### B.2.1 The importance of the $\rho$ update scheme

Our proposed $\rho$ update scheme, theoretically supported by Theorem 1, ensures that *ALPS* converges rapidly while finding high quality solutions. In contrast, ADMM with a fixed $\rho$ may fail to converge when applied to $\ell_0$ constrained least squares problems. To provide empirical evidence for this claim, we compare *ALPS* with ADMM using fixed $\rho$ values. We examined two key metrics: reconstruction loss (objective) and the rate of change of the support (of weights) between consecutive iterations (this measures the convergence speed of the algorithm).Tables 4 and 5 present the results for these metrics, respectively. Our findings show that ADMM with a large $\rho(= 3)$ converges quickly but yields poor solutions, while a small $\rho(= 0.3)$ fails to converge. *ALPS*, utilizing our $\rho$ update scheme, achieves both rapid convergence and high-quality solutions.

| Supp change / Iter | 5 | 10 | 20 | 30 | 50 | 100 |
|---|---|---|---|---|---|---|
| *ALPS* | 1.63e-1 | 1.28e-1 | 5.95e-2 | 5.32e-2 | 5.31e-2 | 5.31e-2 |
| ADMM($\rho = 0.3$) | 7.83e-2 | 7.55e-2 | 7.50e-2 | 7.47e-2 | 7.47e-2 | 7.45e-2 |
| ADMM($\rho = 0.3$) | 9.32e-2 | 8.18e-2 | 7.64e-2 | 7.53e-2 | 7.45e-2 | 7.42e-2 |

Table 4: The relative reconstruction error $\|\mathbf{X}\widehat{\mathbf{W}} - \mathbf{X}\mathbf{W}\|_F^2 / \|\mathbf{X}\widehat{\mathbf{W}}\|_F^2$ over iterations, comparing *ALPS* with ADMM using a fixed penalty parameter $\rho$.

| Supp change / Iter | 5 | 10 | 20 | 30 | 50 | 100 |
|---|---|---|---|---|---|---|
| *ALPS* | 20.2% | 17.0% | 2.8% | 0.0% | 0.0% | 0.0% |
| ADMM($\rho = 0.3$) | 6.4% | 7.0% | 7.0% | 7.0% | 6.9% | 6.9% |
| ADMM($\rho = 0.3$) | 0.2% | < 0.1% | < 0.1% | < 0.1% | < 0.1% | < 0.1% |

Table 5: The rate of change of the support (of weights) between consecutive iterations, comparing *ALPS* with ADMM using a fixed penalty parameter $\rho$.

| Algorithm | OPT-1.3B | OPT-2.7B | OPT-6.7B | OPT-13B | OPT-30B | LLaMA2-7B | LLaMA2-13B | LLaMA3-8B |
|---|---|---|---|---|---|---|---|---|
| MP | 4.7 | 8.9 | 23 | 47 | 120 | 25 | 46 | 24 |
| Wanda | 99 | 161 | 280 | 502 | 1027 | 214 | 407 | 1118 |
| SparseGPT | 363 | 728 | 1621 | 2980 | 6662 | 1263 | 2319 | 2392 |
| DSnoT | 125 | 213 | 417 | 758 | 1528 | 347 | 651 | 1176 |
| *ALPS* | 963 | 2360 | 6069 | 14323 | 48366 | 3043 | 7145 | 6735 |

Table 6: Runtime (in seconds) comparison for one-shot unstructured pruning of OPT models and LLaMA models. Here, runtime includes input activation generation and model pruning.

### B.2.2 Runtime comparison

We compare the runtime of *ALPS* with other methods in Table 6. *ALPS* employs an advanced optimization method to solve the layerwise reconstruction problem, which results in longer running times compared to other algorithms. However, it's important to note that *ALPS* 's runtime is still negligible when compared to fine-tuning methods for LLMs, e.g., LoRA [Hu et al., 2021].

### B.2.3 Comparison of *ALPS* and ADMM-Grad

We discuss the difference between *ALPS* and ADMM-Grad [Boža, 2024], another interesting work using ADMM for LLM unstructured pruning. [Boža, 2024] selects the sparsity mask via iterative magnitude pruning and applies ADMM to solve problem (6) with the selected sparsity mask. *ALPS*, in contrast, is an end-to-end approach that directly targets the $\ell_0$ constrained least squares problem (1). By simultaneously optimizing both weights and sparsity patterns, *ALPS* achieves lower layerwise reconstruction loss compared to ADMM-Grad, as demonstrated in Table 7.

Additionally, since [Boža, 2024] employs ADMM for solving problem (6), we compare it with our proposed PCG procedure. We tested both approaches for solving problem (6) with a given support. Results presented in Table 8 show that PCG outperforms ADMM-Grad in both computational time and objective value. The time advantage of PCG stems from its ability to backsolve without explicitly computing matrix inverses.

| Sparsity | 0.4 | 0.5 | 0.6 | 0.7 | 0.8 | 0.9 |
|---|---|---|---|---|---|---|
| *ALPS* | 3.55e-3 | 7.56e-3 | 1.47e-2 | 2.77e-2 | 5.32e-2 | 1.13e-1 |
| ADMM-Grad | 4.47e-3 | 9.53e-3 | 1.83e-2 | 3.36e-2 | 6.19e-2 | 1.25e-1 |

Table 7: Relative reconstruction error $\|\mathbf{X}\widehat{\mathbf{W}} - \mathbf{X}\mathbf{W}\|_F^2 / \|\mathbf{X}\widehat{\mathbf{W}}\|_F^2$ comparison between *ALPS* and ADMM-Grad across different sparsity levels.

| Sparsity | | 0.4 | 0.5 | 0.6 | 0.7 | 0.8 | 0.9 |
|---|---|---|---|---|---|---|---|
| *ALPS* | time | 0.01s | 0.01s | 0.01s | 0.01s | 0.01s | 0.01s |
| | loss | 3.55e-3 | 7.56e-3 | 1.47e-2 | 2.77e-2 | 5.32e-2 | 1.13e-1 |
| ADMM-Grad | time | 0.11s | 0.11s | 0.11s | 0.11s | 0.11s | 0.11s |
| | loss | 5.1e-3 | 1.12e-2 | 2.32e-2 | 4.49e-2 | 8.83e-2 | 2.00e-1 |

Table 8: Comparison of PCG (Algorithm 2) and ADMM [Boža, 2024] for solving problem (6). Results show runtime (seconds) and layerwise reconstruction loss $\|\mathbf{X}\widehat{\mathbf{W}} - \mathbf{X}\mathbf{W}\|_F^2 / \|\mathbf{X}\widehat{\mathbf{W}}\|_F^2$ across supports with different sparsity levels.

### B.2.4 Pruned model performance on MMLU benchmark

We evaluated the one-shot unstructured pruning performance of MP, SparseGPT, Wanda, DSnoT, and *ALPS* on LLaMA3-8B using the MMLU benchmark to give a comprehensive assessment of each method's effectiveness. Table 1 shows the mean accuracy across all MMLU categories. The results demonstrate that *ALPS* outperforms other methods, further validating its effectiveness in producing high-performance pruned models. We also observe significant performance degradation on MMLU

at high sparsity levels, suggesting that one-shot pruning should be combined with fine-tuning [Hu et al., 2021] or prompting [Sahoo et al., 2024] techniques to maintain model performance in practical applications.

| Sparsity | MP | Wanda | SparseGPT | DSnoT | *ALPS* |
|---|---|---|---|---|---|
| 0.4 | 47.30 | 53.48 | 56.74 | 54.19 | **57.42** |
| 0.5 | 36.11 | 42.64 | 50.87 | 44.49 | **51.01** |
| 0.6 | 25.35 | 29.50 | 37.54 | 28.96 | **40.40** |
| 0.7 | 23.10 | 24.78 | 27.99 | 24.73 | **28.71** |
| 2:4 | 25.30 | 28.24 | **34.27** | 29.25 | 33.98 |
| 4:8 | 28.70 | 32.64 | 38.37 | 34.59 | **41.21** |

Table 9: Performance analysis for one-shot unstructured pruning of LLaMA-3 8B models at various sparsity levels using MMLU benchmark.

### B.2.5   Comprehensive model performance across sparsity levels

We compare the one-shot unstructured pruning performance of MP, SparseGPT, Wanda, DSnoT, and *ALPS* on OPT-1.3B-30B models and LLaMA2-7B, LLaMA2-13B, LLaMA3-8B models in Tables 10-17. The models are pruned at unstructured sparsity levels of $40\%$, $50\%$, $60\%$, $70\%$, $80\%$, and $90\%$, as well as at 2:4 and 4:8 sparsity patterns. We evaluate the perplexity of the pruned models on WikiText2, PTB, and C4 datasets. Additionally, we assess the accuracy of the pruned models on PIQA, LAMBADA, ARC-Easy, and ARC-Challenge. However, LAMBADA accuracy results for the LLaMA model have been omitted since LLaMA models have unsatisfactory performance on this dataset without further modifications. For each combination of model, sparsity, and pruning approach, we run each method five times and report the mean and standard deviation of each performance criterion.

| Model | Algorithm | WikiText2↓ | PTB↓ | C4↓ | LAMBADA↑ | PIQA↑ | ARC-Easy↑ | ARC-Challenge↑ |
|---|---|---|---|---|---|---|---|---|
| OPT-1.3B | MP | 149.7(±0.0) | 131.9(±0.0) | 95.35(±0.00) | 9.24(±0.00) | 63.93(±0.00) | 45.88(±0.00) | 19.71(±0.00) |
| | Wanda | 16.09(±0.24) | 21.13(±0.20) | 16.43(±0.06) | 46.77(±0.46) | 69.68(±0.30) | 54.99(±0.26) | **24.01**(±0.29) |
| | SparseGPT | 16.34(±0.15) | 19.57(±0.23) | 16.91(±0.28) | 46.69(±0.46) | 68.43(±0.10) | 50.05(±0.30) | 23.02(±0.24) |
| | DSnoT | 16.52(±0.11) | 20.61(±0.22) | 16.73(±0.05) | 47.40(±0.45) | 69.49(±0.24) | 54.06(±0.15) | 23.05(±0.26) |
| | *ALPS* | **15.67**(±0.10) | **18.72**(±0.04) | **15.79**(±0.06) | **50.40**(±0.53) | **71.32**(±0.19) | **56.08**(±0.29) | 23.82(±0.28) |
| OPT-2.7B | MP | 21.83(±0.00) | 27.07(±0.00) | 18.48(±0.00) | 40.11(±0.00) | 71.76(±0.00) | 54.92(±0.00) | 24.91(±0.00) |
| | Wanda | 12.95(±0.29) | 19.27(±0.19) | 13.85(±0.03) | 54.59(±0.40) | 72.46(±0.24) | 59.37(±0.37) | 26.06(±0.35) |
| | SparseGPT | **12.61**(±0.09) | 15.96(±0.06) | 13.67(±0.03) | 55.47(±0.40) | 72.47(±0.25) | 59.42(±0.45) | **26.37**(±0.30) |
| | DSnoT | 12.83(±0.10) | 17.96(±0.49) | 13.84(±0.02) | 55.22(±0.37) | 72.50(±0.26) | 59.13(±0.25) | 26.18(±0.36) |
| | *ALPS* | 12.62(±0.03) | **15.78**(±0.02) | **13.57**(±0.04) | **58.73**(±0.36) | **72.88**(±0.05) | **60.20**(±0.21) | 26.08(±0.36) |
| OPT-6.7B | MP | 16.31(±0.00) | 20.55(±0.00) | 16.11(±0.00) | 39.58(±0.00) | 74.37(±0.00) | 61.07(±0.00) | 27.05(±0.00) |
| | Wanda | 10.98(±0.07) | 15.30(±0.35) | 12.24(±0.02) | 64.09(±0.34) | 75.55(±0.09) | 65.00(±0.24) | 29.88(±0.20) |
| | SparseGPT | 10.94(±0.08) | 13.84(±0.15) | 12.08(±0.03) | 65.27(±0.43) | 75.27(±0.21) | 64.54(±0.36) | 29.97(±0.35) |
| | DSnoT | 10.98(±0.07) | 13.90(±0.06) | 12.20(±0.02) | 64.45(±0.40) | 75.23(±0.11) | 64.85(±0.11) | 29.69(±0.22) |
| | *ALPS* | **10.86**(±0.05) | **13.64**(±0.04) | **12.03**(±0.03) | **66.40**(±0.46) | **75.59**(±0.25) | **65.49**(±0.16) | **30.20**(±0.27) |
| OPT-13B | MP | 59.87(±0.00) | 72.90(±0.00) | 43.10(±0.00) | 11.28(±0.00) | 68.06(±0.00) | 52.90(±0.00) | 23.72(±0.00) |
| | Wanda | 10.60(±0.06) | 13.21(±0.05) | 11.65(±0.01) | 62.95(±0.24) | 75.65(±0.18) | 66.26(±0.22) | 32.13(±0.53) |
| | SparseGPT | 10.38(±0.06) | 12.75(±0.02) | 11.47(±0.02) | 64.07(±0.32) | **75.75**(±0.20) | 66.06(±0.20) | 31.55(±0.38) |
| | DSnoT | 10.50(±0.05) | 12.95(±0.03) | 11.59(±0.01) | **64.88**(±0.45) | 75.47(±0.10) | 66.30(±0.24) | 32.08(±0.19) |
| | *ALPS* | **10.32**(±0.04) | **12.65**(±0.02) | **11.39**(±0.02) | 64.23(±0.56) | 75.66(±0.18) | **66.87**(±0.17) | **32.47**(±0.37) |
| OPT-30B | MP | 20.31(±0.00) | 23.74(±0.00) | 16.70(±0.00) | 35.90(±0.00) | 73.56(±0.00) | 61.83(±0.00) | 27.90(±0.00) |
| | Wanda | 9.45(±0.03) | 12.35(±0.04) | 10.97(±0.01) | 66.54(±0.31) | 77.03(±0.15) | 69.71(±0.14) | 33.38(±0.44) |
| | SparseGPT | 9.47(±0.06) | 12.13(±0.01) | 10.85(±0.02) | 67.36(±0.08) | 77.31(±0.14) | 69.84(±0.13) | 33.52(±0.53) |
| | DSnoT | **9.43**(±0.04) | 12.26(±0.03) | 10.95(±0.01) | **67.75**(±0.40) | 76.96(±0.14) | 69.36(±0.21) | 33.79(±0.23) |
| | *ALPS* | 9.47(±0.05) | **12.11**(±0.02) | **10.81**(±0.02) | 67.42(±0.29) | **77.46**(±0.22) | **69.91**(±0.22) | **34.16**(±0.37) |
| LLaMA2-7B | MP | 7.37(±0.00) | 133.6(±0.0) | 9.50(±0.00) | - | 76.22(±0.00) | 66.62(±0.00) | 38.99(±0.00) |
| | Wanda | 6.07(±0.00) | 25.12(±0.06) | 7.63(±0.00) | - | 77.15(±0.07) | 68.40(±0.19) | 40.17(±0.14) |
| | SparseGPT | 6.11(±0.02) | 27.75(±0.57) | 7.63(±0.00) | - | 77.13(±0.26) | 67.81(±0.75) | 39.83(±0.23) |
| | DSnoT | **5.98**(±0.01) | **24.94**(±0.05) | **7.52**(±0.00) | - | 76.76(±0.15) | 67.81(±0.44) | 38.60(±0.27) |
| | *ALPS* | 6.07(±0.02) | 27.14(±0.49) | 7.58(±0.04) | - | **77.21**(±0.47) | **68.74**(±0.69) | **40.29**(±0.57) |
| LLaMA2-13B | MP | 5.50(±0.00) | 40.99(±0.00) | 7.16(±0.00) | - | 77.53(±0.00) | 69.32(±0.00) | 42.58(±0.00) |
| | Wanda | 5.38(±0.01) | 34.11(±0.17) | 6.98(±0.00) | - | **78.96**(±0.12) | **72.69**(±0.13) | **44.08**(±0.15) |
| | SparseGPT | 5.38(±0.01) | 34.47(±0.56) | 6.95(±0.02) | - | 78.59(±0.28) | 71.78(±0.30) | 43.79(±0.30) |
| | DSnoT | **5.30**(±0.00) | **32.40**(±0.12) | **6.88**(±0.00) | - | 78.61(±0.10) | 71.52(±0.36) | 41.84(±0.19) |
| | *ALPS* | 5.36(±0.01) | 33.44(±0.26) | 6.91(±0.03) | - | 78.38(±0.15) | 72.58(±0.30) | 43.70(±0.16) |
| LLaMA3-8B | MP | 20.83(±0.0) | 21.29(±0.0) | 23.69(±0.0) | - | 73.45(±0.0) | 67.09(±0.0) | 38.99(±0.0) |
| | Wanda | 7.45(±0.00) | 12.04(±0.01) | 10.70(±0.00) | - | 77.74(±0.19) | 75.55(±0.53) | 45.90(±0.34) |
| | SparseGPT | 7.52(±0.02) | 11.93(±0.06) | 10.51(±0.08) | - | 78.23(±0.17) | 76.76(±0.61) | **46.83**(±0.58) |
| | DSnoT | **7.43**(±0.00) | 11.96(±0.02) | 10.64(±0.00) | - | 77.58(±0.25) | 74.81(±0.52) | 45.29(±0.58) |
| | *ALPS* | 7.45(±0.02) | **11.68**(±0.03) | **10.31**(±0.11) | - | **78.68**(±0.15) | **76.89**(±0.85) | 46.09(±0.58) |

Table 10: Performance analysis for one-shot unstructured pruning of OPT models and LLaMA models at 40% sparsity. (↑): higher is better; (↓): lower is better. We omit LLaMA results on LAMBADA due to its poor performance without modifications.

| Model | Algorithm | WikiText2↓ | PTB↓ | C4↓ | LAMBADA↑ | PIQA↑ | ARC-Easy↑ | ARC-Challenge↑ |
|---|---|---|---|---|---|---|---|---|
| OPT-1.3B | MP | 799.5(±0.0) | 536.0(±0.0) | 270.6(±0.0) | 1.31(±0.00) | 59.85(±0.00) | 37.12(±0.00) | 18.34(±0.00) |
| | Wanda | 18.62(±0.36) | 26.60(±0.31) | 18.74(±0.14) | 41.10(±0.85) | 68.93(±0.36) | 52.79(±0.48) | 22.46(±0.22) |
| | SparseGPT | 17.77(±0.09) | 22.08(±0.22) | 17.55(±0.22) | 45.82(±1.00) | 68.07(±0.45) | 48.54(±0.47) | 22.59(±0.27) |
| | DSnoT | 19.24(±0.15) | 26.19(±0.63) | 19.12(±0.13) | 41.01(±0.80) | 68.34(±0.19) | 51.49(±0.39) | 21.52(±0.36) |
| | *ALPS* | **17.29**(±0.12) | **21.61**(±0.24) | **16.97**(±0.15) | **47.53**(±1.10) | **70.36**(±0.38) | **54.17**(±0.47) | **23.33**(±0.40) |
| OPT-2.7B | MP | 119.8(±0.0) | 138.7(±0.0) | 55.45(±0.00) | 17.99(±0.00) | 68.61(±0.00) | 46.51(±0.00) | 22.27(±0.00) |
| | Wanda | 14.36(±0.28) | 21.72(±0.09) | 15.09(±0.08) | 49.39(±0.55) | 71.68(±0.33) | 56.91(±0.36) | 24.44(±0.34) |
| | SparseGPT | 13.49(±0.09) | 18.75(±0.49) | 14.39(±0.08) | 53.88(±0.78) | 71.95(±0.41) | 57.45(±0.40) | **25.96**(±0.36) |
| | DSnoT | 14.44(±0.13) | 21.41(±0.24) | 15.18(±0.06) | 51.23(±0.77) | 71.48(±0.20) | 55.36(±0.31) | 24.32(±0.35) |
| | *ALPS* | **13.33**(±0.13) | **17.64**(±0.26) | **14.19**(±0.10) | **57.81**(±0.72) | **72.51**(±0.44) | **59.22**(±0.56) | 25.87(±0.26) |
| OPT-6.7B | MP | 532.3(±0.0) | 222.9(±0.0) | 222.6(±0.0) | 2.92(±0.00) | 67.74(±0.00) | 47.14(±0.00) | 22.35(±0.00) |
| | Wanda | 12.02(±0.11) | 17.72(±0.10) | 13.08(±0.05) | 60.76(±0.55) | 74.48(±0.23) | 63.66(±0.33) | 27.85(±0.34) |
| | SparseGPT | 11.64(±0.09) | 15.85(±0.44) | 12.60(±0.08) | 64.68(±0.66) | 74.69(±0.18) | 63.48(±0.53) | 28.77(±0.42) |
| | DSnoT | 11.97(±0.12) | 15.70(±0.10) | 13.09(±0.05) | 61.33(±0.55) | 73.95(±0.26) | 62.74(±0.36) | 27.61(±0.27) |
| | *ALPS* | **11.46**(±0.09) | **15.26**(±0.28) | **12.45**(±0.08) | **66.31**(±0.57) | **74.86**(±0.18) | **64.67**(±0.35) | **29.16**(±0.52) |
| OPT-13B | MP | 2963(±0) | 2734(±0) | 3240(±0) | 0.00(±0.00) | 57.78(±0.00) | 34.76(±0.00) | 20.05(±0.00) |
| | Wanda | 11.99(±0.09) | 15.95(±0.23) | 12.54(±0.04) | 60.90(±0.71) | 74.71(±0.19) | 63.33(±0.33) | 30.07(±0.18) |
| | SparseGPT | 11.17(±0.10) | 13.91(±0.04) | 11.93(±0.06) | 64.25(±0.91) | 74.67(±0.23) | 65.00(±0.29) | 30.60(±0.17) |
| | DSnoT | 11.53(±0.07) | 14.41(±0.10) | 12.38(±0.03) | 63.38(±0.66) | 74.44(±0.10) | 63.70(±0.46) | 30.36(±0.39) |
| | *ALPS* | **10.76**(±0.07) | **13.48**(±0.06) | **11.71**(±0.06) | **65.21**(±0.48) | **75.11**(±0.10) | **66.09**(±0.20) | **32.15**(±0.79) |
| OPT-30B | MP | 107.8(±0.0) | 245.8(±0.0) | 69.17(±0.00) | 4.99(±0.00) | 65.23(±0.00) | 45.79(±0.00) | 20.56(±0.00) |
| | Wanda | 10.03(±0.06) | 13.37(±0.24) | 11.48(±0.02) | 64.95(±0.66) | 76.77(±0.04) | 68.30(±0.18) | 32.06(±0.24) |
| | SparseGPT | 9.75(±0.05) | 12.86(±0.11) | 11.10(±0.05) | **67.83**(±0.14) | 76.72(±0.17) | **69.54**(±0.32) | 32.88(±0.53) |
| | DSnoT | 10.03(±0.06) | 13.03(±0.02) | 11.44(±0.02) | 67.02(±0.57) | 76.72(±0.15) | 68.18(±0.21) | 32.83(±0.21) |
| | *ALPS* | **9.71**(±0.07) | **12.68**(±0.10) | **11.01**(±0.05) | 67.71(±0.56) | **77.36**(±0.31) | 69.35(±0.20) | **33.77**(±0.61) |
| LLaMA2-7B | MP | 11.82(±0.00) | 1206(±0) | 17.21(±0.00) | - | 74.27(±0.00) | 60.10(±0.00) | 33.62(±0.00) |
| | Wanda | 6.93(±0.01) | **29.89**(±0.05) | 8.54(±0.01) | - | 76.26(±0.22) | 66.14(±0.35) | 36.21(±0.60) |
| | SparseGPT | 7.00(±0.02) | 46.85(±2.55) | 8.46(±0.07) | - | 76.12(±0.28) | 65.07(±1.10) | 36.48(±0.57) |
| | DSnoT | 6.94(±0.02) | 30.25(±0.16) | 8.55(±0.01) | - | 75.47(±0.22) | 66.03(±0.33) | 34.61(±0.46) |
| | *ALPS* | **6.85**(±0.04) | 39.95(±2.61) | **8.26**(±0.10) | - | **76.43**(±0.10) | **66.62**(±0.46) | **37.18**(±0.99) |
| LLaMA2-13B | MP | 6.41(±0.00) | 79.31(±0.00) | 8.23(±0.00) | - | 73.72(±0.00) | 58.71(±0.00) | 35.07(±0.00) |
| | Wanda | 5.98(±0.01) | 40.28(±0.26) | 7.68(±0.01) | - | **78.30**(±0.06) | **71.43**(±0.39) | 40.89(±0.31) |
| | SparseGPT | 6.04(±0.02) | 41.81(±0.36) | 7.62(±0.06) | - | 77.66(±0.26) | 69.27(±0.72) | **40.90**(±0.89) |
| | DSnoT | 5.95(±0.01) | **37.92**(±0.37) | 7.63(±0.01) | - | 77.40(±0.17) | 70.51(±0.48) | 39.16(±0.37) |
| | *ALPS* | **5.95**(±0.04) | 40.33(±0.66) | **7.46**(±0.08) | - | 78.28(±0.15) | 70.17(±0.60) | 40.82(±0.47) |
| LLaMA3-8B | MP | 54.40(±0.0) | 72.28(±0.0) | 69.50(±0.0) | - | 67.90(±0.0) | 60.52(±0.0) | 32.42(±0.0) |
| | Wanda | 9.87(±0.03) | 15.32(±0.02) | 13.49(±0.01) | - | 75.57(±0.32) | 71.19(±0.16) | 40.48(±0.32) |
| | SparseGPT | 9.47(±0.05) | 14.30(±0.10) | 12.35(±0.17) | - | 75.82(±0.39) | 73.72(±0.83) | 41.18(±1.17) |
| | DSnoT | 9.94(±0.06) | 15.22(±0.07) | 13.49(±0.03) | - | 75.44(±0.20) | 71.24(±0.34) | 39.52(±0.20) |
| | *ALPS* | **9.11**(±0.07) | **13.38**(±0.08) | **11.78**(±0.23) | - | **76.90**(±0.23) | **75.15**(±0.48) | **42.99**(±1.26) |

Table 11: Performance analysis for one-shot unstructured pruning of OPT models and LLaMA models at 50% sparsity. (↑): higher is better; (↓): lower is better. We omit LLaMA results on LAMBADA due to its poor performance without modifications.

| Model | Algorithm | WikiText2 ↓ | PTB ↓ | C4 ↓ | LAMBADA ↑ | PIQA ↑ | ARC-Easy ↑ | ARC-Challenge ↑ |
|---|---|---|---|---|---|---|---|---|
| OPT-1.3B | MP | 6136(±0) | 4017(±0) | 2232(±0) | 0.00(±0.00) | 55.60(±0.00) | 30.35(±0.00) | 19.20(±0.00) |
|  | Wanda | 27.17(±0.20) | 42.55(±0.28) | 25.46(±0.34) | 28.94(±0.98) | 66.05(±0.27) | 47.95(±0.44) | 21.31(±0.29) |
|  | SparseGPT | 22.86(±0.30) | 29.29(±0.17) | 20.94(±0.33) | 39.80(±1.26) | 65.47(±0.64) | 46.14(±0.82) | 21.43(±0.41) |
|  | DSnoT | 30.47(±0.34) | 50.08(±0.69) | 29.38(±0.41) | 27.86(±0.56) | 63.85(±0.49) | 44.76(±0.39) | 20.84(±0.46) |
|  | *ALPS* | **21.51**(±0.53) | **28.17**(±0.50) | **19.62**(±0.40) | **43.54**(±1.15) | **67.53**(±0.32) | **51.00**(±0.94) | **22.53**(±0.44) |
| OPT-2.7B | MP | 4283(±0) | 3504(±0) | 2229(±0) | 0.04(±0.00) | 57.67(±0.00) | 34.60(±0.00) | 20.22(±0.00) |
|  | Wanda | 20.24(±0.28) | 31.78(±0.35) | 19.74(±0.20) | 34.27(±0.87) | 69.88(±0.21) | 52.37(±0.58) | 22.18(±0.49) |
|  | SparseGPT | 16.21(±0.18) | 24.00(±0.29) | 16.24(±0.20) | 48.59(±0.88) | 70.79(±0.12) | 55.08(±0.17) | 24.28(±0.40) |
|  | DSnoT | 20.72(±0.25) | 33.01(±0.34) | 20.32(±0.16) | 38.07(±1.56) | 68.53(±0.24) | 50.76(±0.49) | 21.76(±0.42) |
|  | *ALPS* | **15.67**(±0.20) | **22.06**(±0.26) | **15.75**(±0.26) | **53.43**(±0.91) | **71.34**(±0.29) | **56.09**(±0.16) | **25.14**(±0.55) |
| OPT-6.7B | MP | 9643(±0) | 4791(±0) | 5876(±0) | 0.00(±0.00) | 53.86(±0.00) | 26.30(±0.00) | 20.82(±0.00) |
|  | Wanda | 15.34(±0.19) | 24.50(±0.18) | 16.17(±0.42) | 49.10(±1.49) | 72.50(±0.28) | 58.27(±0.48) | 25.26(±0.22) |
|  | SparseGPT | 13.53(±0.21) | 18.78(±0.20) | 13.92(±0.20) | 60.60(±0.98) | 73.39(±0.30) | 61.64(±0.33) | 27.10(±0.42) |
|  | DSnoT | 15.32(±0.19) | 23.30(±0.21) | 16.00(±0.13) | 51.53(±1.28) | 71.86(±0.27) | 57.98(±0.11) | 24.85(±0.56) |
|  | *ALPS* | **13.05**(±0.14) | **17.76**(±0.25) | **13.52**(±0.23) | **64.00**(±1.48) | **74.03**(±0.29) | **62.46**(±0.40) | **27.24**(±0.86) |
| OPT-13B | MP | 111710(±0) | 24830(±0) | 14398(±0) | 0.00(±0.00) | 52.61(±0.00) | 26.18(±0.00) | 21.59(±0.00) |
|  | Wanda | 16.09(±0.13) | 22.45(±0.25) | 15.38(±0.10) | 50.93(±1.39) | 71.55(±0.26) | 58.78(±0.15) | 28.38(±0.41) |
|  | SparseGPT | 13.04(±0.09) | 17.32(±0.09) | 13.11(±0.16) | 61.96(±1.00) | 73.14(±0.30) | 61.15(±0.69) | 27.97(±0.69) |
|  | DSnoT | 15.11(±0.14) | 20.03(±0.42) | 15.10(±0.09) | 54.96(±1.24) | 71.88(±0.26) | 58.49(±0.24) | 27.35(±0.18) |
|  | *ALPS* | **12.05**(±0.18) | **15.77**(±0.16) | **12.55**(±0.16) | **64.58**(±1.32) | **74.19**(±0.32) | **64.05**(±0.43) | **30.07**(±0.60) |
| OPT-30B | MP | 13224(±0) | 6381(±0) | 6069(±0) | 0.00(±0.00) | 53.54(±0.00) | 26.26(±0.00) | 19.80(±0.00) |
|  | Wanda | 24.85(±1.97) | 25.88(±1.78) | 21.13(±1.09) | 38.37(±2.09) | 74.60(±0.47) | 64.52(±0.48) | 29.04(±0.30) |
|  | SparseGPT | 10.67(±0.07) | 14.61(±0.20) | 11.78(±0.12) | 67.26(±0.72) | 76.00(±0.45) | 67.63(±0.23) | **32.63**(±0.81) |
|  | DSnoT | 14.88(±0.26) | 18.27(±0.33) | 15.58(±0.24) | 56.47(±1.05) | 74.77(±0.23) | 65.27(±0.33) | 30.27(±0.24) |
|  | *ALPS* | **10.43**(±0.09) | **14.23**(±0.15) | **11.52**(±0.12) | **67.66**(±0.73) | **76.34**(±0.30) | **68.09**(±0.38) | 32.58(±0.67) |
| LLaMA2-7B | MP | 7058(±0) | 285318(±0) | 42050(±0) | - | 63.11(±0.00) | 44.99(±0.00) | 26.37(±0.00) |
|  | Wanda | 10.82(±0.04) | 65.23(±0.84) | 12.61(±0.03) | - | 71.19(±0.18) | 59.24(±0.48) | 30.56(±0.63) |
|  | SparseGPT | 10.19(±0.06) | 215.8(±47.2) | 11.52(±0.23) | - | 71.78(±0.52) | 58.87(±0.65) | 30.27(±0.66) |
|  | DSnoT | 11.47(±0.09) | **64.02**(±1.18) | 13.41(±0.05) | - | 70.05(±0.19) | 59.00(±0.54) | 28.48(±0.59) |
|  | *ALPS* | **9.36**(±0.19) | 107.6(±14.2) | **10.26**(±0.34) | - | **73.68**(±0.14) | **60.56**(±2.13) | **31.55**(±0.96) |
| LLaMA2-13B | MP | 10.10(±0.00) | 301.5(±0.0) | 13.25(±0.00) | - | 68.17(±0.00) | 46.51(±0.00) | 26.45(±0.00) |
|  | Wanda | 8.44(±0.03) | 89.78(±0.60) | 10.44(±0.03) | - | 75.03(±0.13) | 63.43(±0.56) | 36.48(±0.50) |
|  | SparseGPT | 8.25(±0.10) | 83.82(±3.18) | 9.78(±0.22) | - | 74.95(±0.13) | 64.30(±1.32) | 35.39(±0.57) |
|  | DSnoT | 8.88(±0.04) | 79.41(±1.33) | 10.92(±0.04) | - | 74.09(±0.24) | 63.48(±0.39) | 34.71(±0.32) |
|  | *ALPS* | **7.55**(±0.10) | **65.86**(±3.81) | **8.93**(±0.26) | - | **76.16**(±0.34) | **67.74**(±1.24) | **37.66**(±0.78) |
| LLaMA3-8B | MP | 410664(±0.0) | 317003(±0.0) | 66461(±0.0) | - | 62.68(±0.0) | 39.23(±0.0) | 23.38(±0.0) |
|  | Wanda | 23.55(±0.24) | 37.74(±0.57) | 29.55(±0.16) | - | 68.30(±0.19) | 60.04(±0.27) | 27.85(±0.42) |
|  | SparseGPT | 15.63(±0.14) | 22.89(±0.58) | 17.92(±0.50) | - | 71.81(±0.55) | 63.20(±0.85) | 31.96(±1.08) |
|  | DSnoT | 23.05(±0.16) | 36.15(±0.40) | 27.71(±0.17) | - | 67.85(±0.49) | 60.51(±0.30) | 28.07(±0.67) |
|  | *ALPS* | **14.09**(±0.18) | **21.26**(±0.69) | **15.67**(±0.77) | - | **73.41**(±0.64) | **67.10**(±0.57) | **34.52**(±0.83) |

Table 12: Performance analysis for one-shot unstructured pruning of OPT models and LLaMA models at 60% sparsity. (↑): higher is better; (↓): lower is better. We omit LLaMA results on LAMBADA due to its poor performance without modifications.

| Model | Algorithm | WikiText2 ↓ | PTB ↓ | C4 ↓ | LAMBADA ↑ | PIQA ↑ | ARC-Easy ↑ | ARC-Challenge ↑ |
|---|---|---|---|---|---|---|---|---|
| OPT-1.3B | MP | 9409(±0) | 6689(±0) | 5652(±0) | 0.00(±0.00) | 52.12(±0.00) | 26.18(±0.00) | 20.90(±0.00) |
|  | Wanda | 100.8(±3.8) | 128.4(±3.1) | 78.58(±2.34) | 9.77(±0.63) | 59.38(±0.46) | 37.92(±0.32) | 18.02(±0.36) |
|  | SparseGPT | 52.02(±2.22) | 70.10(±3.31) | 37.05(±1.78) | 27.27(±1.04) | 62.38(±0.79) | 40.79(±0.58) | 19.69(±0.78) |
|  | DSnoT | 367.5(±19.6) | 370.2(±30.6) | 205.4(±6.1) | 8.62(±0.23) | 56.90(±0.37) | 33.27(±0.47) | 17.49(±0.57) |
|  | *ALPS* | **39.50**(±2.38) | **50.68**(±1.13) | **28.52**(±1.25) | **32.11**(±1.71) | **64.43**(±0.37) | **45.01**(±1.03) | **21.08**(±0.32) |
| OPT-2.7B | MP | 12249(±0) | 10993(±0) | 9960(±0) | 0.00(±0.00) | 52.94(±0.00) | 26.52(±0.00) | 19.71(±0.00) |
|  | Wanda | 365.1(±16.5) | 379.4(±37.6) | 224.4(±4.8) | 5.02(±0.40) | 58.01(±0.38) | 34.85(±0.38) | 17.61(±0.28) |
|  | SparseGPT | 28.93(±1.62) | 40.89(±1.19) | 23.11(±0.66) | 34.96(±1.97) | 66.50(±0.32) | 49.55(±0.50) | 21.67(±0.57) |
|  | DSnoT | 114.8(±1.5) | 116.2(±6.7) | 75.28(±3.51) | 9.39(±0.63) | 59.62(±0.23) | 37.37(±0.48) | 18.43(±0.78) |
|  | *ALPS* | **25.36**(±1.34) | **35.76**(±0.90) | **20.93**(±0.75) | **42.53**(±2.40) | **67.62**(±0.36) | **51.55**(±0.84) | **22.27**(±0.43) |
| OPT-6.7B | MP | 9970(±0) | 4779(±0) | 5055(±0) | 0.00(±0.00) | 52.67(±0.00) | 26.60(±0.00) | 21.16(±0.00) |
|  | Wanda | 162.9(±8.7) | 204.9(±10.6) | 206.0(±13.5) | 2.82(±0.10) | 58.13(±0.16) | 35.82(±0.43) | 17.22(±0.39) |
|  | SparseGPT | 21.14(±0.69) | 29.34(±0.44) | 19.07(±0.62) | 47.20(±0.94) | 69.23(±0.76) | 54.67(±0.20) | 24.08(±0.28) |
|  | DSnoT | 7985(±465) | 6572(±783) | 4764(±478) | 0.15(±0.12) | 53.19(±0.36) | 29.13(±0.78) | 18.46(±0.48) |
|  | *ALPS* | **18.99**(±0.85) | **24.89**(±0.26) | **17.01**(±0.60) | **56.04**(±1.32) | **71.39**(±0.27) | **58.52**(±0.74) | **26.19**(±0.52) |
| OPT-13B | MP | 524559(±0) | 146680(±0) | 155160(±0) | 0.00(±0.00) | 52.99(±0.00) | 25.25(±0.00) | 22.95(±0.00) |
|  | Wanda | 63.42(±2.22) | 66.50(±1.93) | 50.19(±2.28) | 10.82(±0.95) | 63.03(±0.72) | 43.66(±0.52) | 23.50(±0.63) |
|  | SparseGPT | 19.29(±0.42) | 25.50(±0.32) | 16.79(±0.45) | 47.97(±1.73) | 69.16(±0.31) | 53.51(±0.49) | 26.23(±0.96) |
|  | DSnoT | 78.13(±2.94) | 60.11(±0.91) | 56.39(±1.79) | 14.43(±0.41) | 61.57(±0.60) | 40.52(±0.62) | 20.77(±0.67) |
|  | *ALPS* | **16.71**(±0.62) | **21.49**(±0.40) | **15.07**(±0.46) | **57.08**(±1.62) | **71.73**(±0.22) | **59.30**(±0.50) | **27.97**(±0.93) |
| OPT-30B | MP | 26271(±0) | 12693(±0) | 13057(±0) | 0.00(±0.00) | 52.23(±0.00) | 25.46(±0.00) | 19.97(±0.00) |
|  | Wanda | 10384(±198) | 5467(±104) | 6692(±191) | 0.01(±0.01) | 52.21(±0.27) | 25.61(±0.10) | 20.20(±0.41) |
|  | SparseGPT | 13.61(±0.22) | 18.94(±0.25) | 13.96(±0.35) | 60.50(±0.88) | 73.74(±0.27) | 63.26(±0.44) | 28.81(±0.40) |
|  | DSnoT | 11328(±368) | 5685(±248) | 6579(±298) | 0.04(±0.02) | 52.48(±0.30) | 26.36(±0.15) | 20.15(±0.31) |
|  | *ALPS* | **12.67**(±0.25) | **17.47**(±0.27) | **13.00**(±0.32) | **63.52**(±1.38) | **75.05**(±0.38) | **65.52**(±0.75) | **30.55**(±0.76) |
| LLaMA2-7B | MP | 141884(±0) | 56430(±0) | 9843(±0) | - | 52.61(±0.00) | 28.24(±0.00) | 21.33(±0.00) |
|  | Wanda | 76.05(±1.45) | **320.0**(±6.3) | 68.45(±2.19) | - | 55.59(±0.43) | 28.86(±0.27) | 18.74(±0.55) |
|  | SparseGPT | 27.20(±0.75) | 1049(±256) | 26.14(±0.92) | - | 62.76(±0.34) | 39.62(±0.83) | 22.47(±0.63) |
|  | DSnoT | 63.12(±2.82) | 322.0(±9.7) | 61.91(±2.05) | - | 56.30(±0.25) | 29.72(±0.17) | 19.06(±0.20) |
|  | *ALPS* | **19.31**(±0.68) | 480.1(±198.7) | **17.66**(±1.03) | - | **66.82**(±0.92) | **48.37**(±0.31) | **24.95**(±0.64) |
| LLaMA2-13B | MP | 190.9(±0.0) | 2246(±0) | 192.1(±0.0) | - | 60.45(±0.00) | 35.19(±0.00) | 20.73(±0.00) |
|  | Wanda | 47.20(±1.52) | 410.3(±3.6) | 47.03(±1.10) | - | 58.39(±0.35) | 35.62(±0.35) | 18.67(±0.36) |
|  | SparseGPT | 19.93(±0.66) | 275.6(±16.9) | 20.08(±0.84) | - | 66.14(±0.41) | 48.27(±0.31) | 26.31(±0.53) |
|  | DSnoT | 47.21(±0.79) | 328.9(±4.8) | 47.93(±0.83) | - | 57.19(±0.23) | 33.05(±0.32) | 18.24(±0.42) |
|  | *ALPS* | **14.23**(±0.43) | **152.8**(±6.9) | **14.14**(±0.88) | - | **70.21**(±0.70) | **56.48**(±0.48) | **29.39**(±0.53) |
| LLaMA3-8B | MP | 572874(±0.0) | 243236(±0.0) | 211703(±0.0) | - | 52.77(±0.0) | 26.64(±0.0) | 21.50(±0.0) |
|  | Wanda | 116.8(±3.1) | 174.8(±8.8) | 121.2(±4.0) | - | 55.75(±0.43) | 32.02(±0.43) | 17.94(±0.19) |
|  | SparseGPT | 43.15(±2.62) | 76.50(±5.28) | 38.46(±2.43) | - | 62.64(±1.20) | 44.88(±1.99) | 21.67(±0.87) |
|  | DSnoT | 108.7(±2.3) | 166.9(±4.0) | 114.2(±1.2) | - | 55.50(±0.29) | 31.60(±0.10) | 17.78(±0.13) |
|  | *ALPS* | **30.53**(±1.58) | **53.12**(±3.28) | **27.42**(±1.89) | - | **65.15**(±0.92) | **50.61**(±0.96) | **23.23**(±1.07) |

Table 13: Performance analysis for one-shot unstructured pruning of OPT models and LLaMA models at 70% sparsity. (↑): higher is better; (↓): lower is better. We omit LLaMA results on LAMBADA due to its poor performance without modifications.

| Model | Algorithm | WikiText2 ↓ | PTB ↓ | C4 ↓ | LAMBADA ↑ | PIQA ↑ | ARC-Easy ↑ | ARC-Challenge ↑ |
|---|---|---|---|---|---|---|---|---|
| OPT-1.3B | MP | 10988(±0) | 8966(±0) | 7767(±0) | 0.00(±0.00) | 52.99(±0.00) | 24.58(±0.00) | **19.62**(±0.00) |
| | Wanda | 3327(±412) | 1783(±169) | 1306(±145) | 0.32(±0.05) | 55.17(±0.38) | 28.80(±0.30) | 18.46(±0.65) |
| | SparseGPT | 1005(±76) | 521.0(±36.2) | 311.9(±27.4) | 5.91(±0.47) | 55.59(±0.52) | 29.81(±0.34) | 18.92(±0.74) |
| | DSnoT | 10757(±798) | 7749(±509) | 5559(±287) | 0.00(±0.00) | 53.84(±0.20) | 26.03(±0.32) | 18.79(±0.53) |
| | *ALPS* | **326.7**(±63.3) | **311.3**(±38.5) | **117.3**(±13.3) | **11.38**(±0.55) | **57.29**(±0.80) | **32.11**(±0.69) | 18.26(±0.43) |
| OPT-2.7B | MP | 19200(±0) | 9475(±0) | 13627(±0) | 0.00(±0.00) | 52.45(±0.00) | 26.01(±0.00) | 20.65(±0.00) |
| | Wanda | 6580(±705) | 4946(±601) | 4861(±763) | 0.00(±0.00) | 53.45(±0.68) | 27.06(±0.29) | 20.44(±0.47) |
| | SparseGPT | 154.9(±11.2) | 142.7(±8.4) | 73.43(±4.16) | 12.02(±0.53) | 58.05(±0.39) | 34.88(±0.54) | 18.63(±0.34) |
| | DSnoT | 18322(±2378) | 11516(±1479) | 15210(±2139) | 0.00(±0.00) | 52.10(±0.61) | 26.37(±0.32) | **21.31**(±0.27) |
| | *ALPS* | **108.2**(±7.2) | **114.4**(±2.0) | **53.83**(±3.32) | **19.48**(±0.60) | **59.61**(±0.68) | **38.25**(±0.63) | 19.27(±0.50) |
| OPT-6.7B | MP | 42719(±0) | 18213(±0) | 20049(±0) | 0.00(±0.00) | 52.45(±0.00) | 26.39(±0.00) | **21.50**(±0.00) |
| | Wanda | 4317(±293) | 2429(±190) | 2321(±189) | 0.17(±0.06) | 54.27(±0.63) | 29.04(±0.43) | 18.28(±0.37) |
| | SparseGPT | 109.1(±5.6) | 92.51(±2.90) | 56.94(±3.22) | 14.28(±0.78) | 60.55(±0.53) | 38.75(±0.97) | 18.87(±0.20) |
| | DSnoT | 10990(±2902) | 8165(±2179) | 8405(±2379) | 0.00(±0.00) | 53.13(±0.35) | 26.46(±0.75) | 20.17(±0.31) |
| | *ALPS* | **70.97**(±5.76) | **73.94**(±3.62) | **40.33**(±2.42) | **25.85**(±1.35) | **62.44**(±0.51) | **44.23**(±0.80) | 20.14(±0.38) |
| OPT-13B | MP | 266874(±0) | 98067(±0) | 97073(±0) | 0.00(±0.00) | 51.74(±0.00) | 25.55(±0.00) | 21.25(±0.00) |
| | Wanda | 11025(±1789) | 4900(±673) | 6256(±897) | 0.00(±0.01) | 53.67(±0.62) | 26.28(±0.79) | 21.52(±0.47) |
| | SparseGPT | 78.05(±6.82) | 72.64(±3.30) | 41.60(±3.02) | 14.56(±0.35) | 62.10(±0.64) | 40.26(±1.19) | 21.71(±0.74) |
| | DSnoT | 11697(±490) | 5860(±320) | 6295(±245) | 0.00(±0.00) | 52.81(±0.32) | 25.72(±0.20) | 21.84(±0.39) |
| | *ALPS* | **42.92**(±3.03) | **45.19**(±1.16) | **27.17**(±1.44) | **31.38**(±1.71) | **65.57**(±0.72) | **48.50**(±0.51) | **22.54**(±0.56) |
| OPT-30B | MP | 68520(±0) | 31584(±0) | 31793(±0) | 0.00(±0.00) | 52.07(±0.00) | 25.97(±0.00) | 20.39(±0.00) |
| | Wanda | 8184(±379) | 3855(±238) | 5387(±382) | 0.00(±0.00) | 51.98(±0.35) | 26.22(±0.36) | 20.07(±0.15) |
| | SparseGPT | 50.33(±5.81) | 48.59(±1.04) | 33.42(±2.83) | 19.79(±1.35) | 63.69(±0.56) | 46.17(±0.48) | 21.91(±0.91) |
| | DSnoT | 11788(±1611) | 5178(±636) | 6403(±814) | 0.00(±0.00) | 51.81(±0.28) | 26.41(±0.26) | 20.12(±0.39) |
| | *ALPS* | **25.05**(±1.84) | **30.64**(±0.57) | **19.29**(±1.14) | **44.46**(±2.43) | **69.72**(±0.61) | **56.65**(±0.59) | **24.98**(±0.39) |
| LLaMA2-7B | MP | 60361(±0) | 72505(±0) | 47003(±0) | - | 53.32(±0.00) | 26.05(±0.00) | **22.70**(±0.00) |
| | Wanda | 3140(±692) | 3158(±669) | 2146(±478) | - | 52.97(±0.36) | 26.20(±0.20) | 20.65(±0.34) |
| | SparseGPT | 114.9(±6.0) | 1733(±68) | 83.04(±3.31) | - | 54.86(±0.59) | 26.34(±0.52) | 19.15(±0.33) |
| | DSnoT | 7300(±648) | 6036(±650) | 6385(±738) | - | 53.05(±0.44) | 26.31(±0.26) | 19.97(±0.47) |
| | *ALPS* | **52.02**(±4.85) | **1462**(±294) | **39.37**(±2.22) | - | **58.05**(±0.45) | **30.27**(±0.59) | 19.97(±0.16) |
| LLaMA2-13B | MP | 12018(±0) | 8573(±0) | 11705(±0) | - | 53.37(±0.00) | 26.81(±0.00) | 21.25(±0.00) |
| | Wanda | 1123(±128) | 4111(±281) | 799.4(±57.9) | - | 53.26(±0.60) | 26.24(±0.21) | 20.44(±0.41) |
| | SparseGPT | 100.8(±4.0) | 605.4(±36.4) | 64.61(±2.25) | - | 54.94(±0.53) | 28.17(±0.40) | 19.37(±0.26) |
| | DSnoT | 13908(±535) | 10634(±842) | 13102(±935) | - | 52.51(±0.30) | 26.73(±0.38) | **22.30**(±0.11) |
| | *ALPS* | **39.07**(±2.83) | **345.8**(±27.3) | **30.70**(±1.85) | - | **59.39**(±0.63) | **34.19**(±0.47) | 21.01(±0.32) |
| LLaMA3-8B | MP | 491419(±0.0) | 283948(±0.0) | 624818(±0.0) | - | 52.77(±0.00) | 24.87(±0.0) | **22.61**(±0.0) |
| | Wanda | 1133(±219) | 1785(±408) | 696.3(±105.4) | - | 52.89(±0.42) | 28.27(±0.60) | 20.12(±0.25) |
| | SparseGPT | 210.0(±19.3) | 342.3(±15.0) | 108.6(±5.4) | - | 54.94(±0.65) | 29.87(±0.37) | 18.46(±0.58) |
| | DSnoT | 959.6(±77.2) | 1061(±58) | 454.0(±17.4) | - | 52.74(±0.20) | 28.01(±0.20) | 19.69(±0.48) |
| | *ALPS* | **120.6**(±7.1) | **158.9**(±10.0) | **61.08**(±3.38) | - | **57.19**(±0.48) | **33.35**(±0.40) | 16.86(±0.58) |

Table 14: Performance analysis for one-shot unstructured pruning of OPT models and LLaMA models at 80% sparsity. (↑): higher is better; (↓): lower is better. We omit LLaMA results on LAMBADA due to its poor performance without modifications.

| Model | Algorithm | WikiText2 ↓ | PTB ↓ | C4 ↓ | LAMBADA ↑ | PIQA ↑ | ARC-Easy ↑ | ARC-Challenge ↑ |
|---|---|---|---|---|---|---|---|---|
| OPT-1.3B | MP | 20486(±0) | 13134(±0) | 11298(±0) | 0.00(±0.00) | 52.99(±0.00) | 25.55(±0.00) | 19.28(±0.00) |
| | Wanda | 13290(±3657) | 7120(±2742) | 7376(±2527) | 0.00(±0.00) | **53.19**(±0.49) | 25.41(±0.54) | **20.15**(±1.16) |
| | SparseGPT | 7771(±589) | 5797(±585) | 2761(±218) | 0.00(±0.01) | 53.00(±0.24) | **25.72**(±0.31) | 19.20(±0.32) |
| | DSnoT | 18519(±3418) | 14488(±2478) | 15067(±2433) | 0.00(±0.00) | 52.67(±0.27) | 24.72(±0.59) | 19.97(±0.77) |
| | *ALPS* | **4891**(±344) | **2900**(±62) | **1415**(±78) | **0.10**(±0.02) | 52.72(±0.58) | 25.66(±0.52) | 19.85(±0.91) |
| OPT-2.7B | MP | 16397(±0) | 11205(±0) | 15578(±0) | 0.00(±0.00) | 52.67(±0.00) | 25.25(±0.00) | **23.89**(±0.00) |
| | Wanda | 15018(±2265) | 11282(±1736) | 9191(±1993) | 0.00(±0.00) | 52.93(±0.92) | 26.10(±0.41) | 21.01(±0.62) |
| | SparseGPT | 5923(±544) | 4546(±416) | 2486(±135) | 0.02(±0.02) | 52.27(±0.42) | **26.68**(±0.16) | 20.36(±0.65) |
| | DSnoT | 18201(±14215) | 13289(±10085) | 15652(±11455) | 0.00(±0.00) | 52.60(±0.58) | 25.40(±0.58) | 22.34(±0.85) |
| | *ALPS* | **3686**(±266) | **1599**(±121) | **800.0**(±47.0) | **0.23**(±0.07) | 52.64(±0.42) | 26.55(±0.56) | 19.93(±0.33) |
| OPT-6.7B | MP | 232525(±0) | 95928(±0) | 115340(±0) | 0.00(±0.00) | 51.90(±0.00) | **26.47**(±0.00) | 20.14(±0.00) |
| | Wanda | 17429(±1621) | 11334(±1067) | 13656(±1433) | 0.00(±0.00) | 52.84(±0.18) | 25.70(±0.56) | 19.42(±0.74) |
| | SparseGPT | 10287(±1598) | 5432(±1322) | 4930(±1756) | 0.00(±0.00) | 52.48(±0.64) | 25.87(±0.47) | 20.15(±0.54) |
| | DSnoT | 11233(±4770) | 7810(±2983) | 9622(±4536) | 0.00(±0.00) | **53.12**(±0.84) | 25.73(±0.43) | 20.44(±0.38) |
| | *ALPS* | **6424**(±194) | **4112**(±154) | **2806**(±174) | **0.01**(±0.02) | 52.35(±0.85) | 26.30(±0.34) | **20.77**(±0.31) |
| OPT-13B | MP | 567627(±0) | 238209(±0) | 202496(±0) | 0.00(±0.00) | 52.29(±0.00) | **26.52**(±0.00) | 21.33(±0.00) |
| | Wanda | 48747(±19963) | 24633(±8918) | 30681(±12658) | 0.00(±0.00) | 52.25(±0.40) | 25.24(±0.43) | 20.09(±0.42) |
| | SparseGPT | 44328(±35324) | 84805(±93235) | 49471(±49558) | 0.00(±0.00) | 51.08(±0.87) | 26.48(±0.67) | 21.26(±0.94) |
| | DSnoT | 40268(±9573) | **18442**(±5345) | 19745(±4020) | 0.00(±0.00) | 52.11(±0.45) | 25.32(±0.45) | **21.88**(±0.77) |
| | *ALPS* | **15020**(±4799) | 43428(±33717) | **10101**(±2617) | **0.02**(±0.03) | **52.50**(±0.30) | 26.23(±0.56) | 20.51(±0.51) |
| OPT-30B | MP | 8458877440(±0) | 2651165184(±0) | 2297142016(±0) | 0.00(±0.00) | 52.07(±0.00) | 26.39(±0.00) | 20.39(±0.00) |
| | Wanda | 11122(±1998) | 6063(±1105) | 7205(±1107) | 0.00(±0.00) | 51.75(±0.30) | 26.06(±0.37) | 20.03(±0.29) |
| | SparseGPT | 8695(±1067) | 4986(±521) | 4380(±537) | 0.00(±0.00) | 52.50(±0.73) | 26.16(±0.35) | **20.89**(±0.56) |
| | DSnoT | 10001(±1011) | 4357(±399) | 5683(±546) | 0.00(±0.00) | 52.27(±0.22) | 26.09(±0.42) | 20.14(±0.20) |
| | *ALPS* | **2353**(±233) | **1192**(±33) | **640.3**(±89.9) | **0.69**(±0.12) | **53.68**(±0.44) | **29.25**(±0.46) | 19.54(±0.83) |
| LLaMA2-7B | MP | 78367(±0) | 160179(±0) | 37324(±0) | - | 53.59(±0.00) | 26.30(±0.00) | **24.91**(±0.00) |
| | Wanda | 10820(±2403) | 7850(±1479) | 8309(±1844) | - | 51.95(±0.55) | **26.46**(±0.35) | 21.43(±0.57) |
| | SparseGPT | 1500(±111) | - | 829.3(±49.9) | - | 52.69(±0.42) | 25.32(±0.39) | 21.55(±0.38) |
| | DSnoT | 13944(±1025) | 21231(±1038) | 15297(±1305) | - | 52.38(±0.36) | 26.36(±0.41) | 21.64(±0.39) |
| | *ALPS* | **228.0**(±11.7) | **1800**(±373) | **135.8**(±7.1) | - | **53.81**(±0.48) | 26.16(±0.25) | 19.66(±0.21) |
| LLaMA2-13B | MP | 201060(±0) | 381074(±0) | 189457(±0) | - | 51.74(±0.00) | 24.92(±0.00) | 21.76(±0.00) |
| | Wanda | 17470(±5922) | 19851(±5184) | 10680(±3894) | - | 52.60(±0.28) | 25.80(±0.50) | 21.13(±0.82) |
| | SparseGPT | 1443(±136) | 2651(±482) | 878.3(±101.7) | - | 53.31(±0.26) | 26.41(±0.26) | 21.35(±0.40) |
| | DSnoT | 46515(±33563) | 26995(±8616) | 27673(±14262) | - | 51.85(±0.18) | 26.04(±0.25) | **21.91**(±0.28) |
| | *ALPS* | **251.1**(±19.6) | **795.9**(±46.7) | **124.2**(±11.7) | - | **53.62**(±0.63) | **26.47**(±0.38) | 19.06(±0.30) |
| LLaMA3-8B | MP | 2115317(±0.0) | 1330667(±0.0) | 1355430(±0.0) | - | 52.50(±0.0) | 24.20(±0.0) | **21.67**(±0.0) |
| | Wanda | 9913(±1033) | 12855(±2742) | 6487(±693) | - | 52.33(±0.45) | 26.52(±0.33) | 20.73(±0.69) |
| | SparseGPT | 1102(±63) | 1641(±121) | 464.5(±22.4) | - | 52.76(±0.33) | 27.52(±0.49) | 19.20(±0.65) |
| | DSnoT | 35323(±9006) | 25843(±5941) | 14828(±3567) | - | 53.06(±0.57) | 26.09(±0.28) | 20.89(±0.38) |
| | *ALPS* | **515.9**(±34.2) | **524.6**(±30.9) | **212.2**(±10.2) | - | **54.47**(±0.54) | **28.89**(±0.62) | 17.65(±0.69) |

Table 15: Performance analysis for one-shot unstructured pruning of OPT models and LLaMA models at 90% sparsity. (↑): higher is better; (↓): lower is better. We omit LLaMA results on LAMBADA due to its poor performance without modifications.

| Model | Algorithm | WikiText2↓ | PTB↓ | C4↓ | LAMBADA↑ | PIQA↑ | ARC-Easy↑ | ARC-Challenge↑ |
|---|---|---|---|---|---|---|---|---|
| OPT-1.3B | MP | 427.1(±0.0) | 343.6(±0.0) | 127.0(±0.0) | 2.38(±0.00) | 61.53(±0.00) | 39.35(±0.00) | 17.75(±0.00) |
| | Wanda | 28.34(±0.37) | 41.67(±0.72) | 25.73(±0.25) | 27.93(±0.70) | 65.73(±0.34) | 47.98(±0.46) | 21.02(±0.42) |
| | SparseGPT | 24.39(±0.62) | 32.50(±1.78) | 22.81(±0.78) | 36.37(±0.67) | 64.73(±0.51) | 45.34(±0.82) | 20.70(±0.34) |
| | DSnoT | 32.94(±0.71) | 50.37(±1.11) | 30.04(±0.35) | 29.97(±0.64) | 64.34(±0.27) | 45.70(±0.32) | 20.75(±0.34) |
| | *ALPS* | **22.68**(±0.58) | **27.63**(±0.25) | **20.16**(±0.36) | **39.86**(±1.11) | **67.15**(±0.77) | **50.11**(±0.69) | **21.60**(±0.43) |
| OPT-2.7B | MP | 1153(±0) | 975.3(±0.0) | 718.6(±0.0) | 0.53(±0.00) | 57.73(±0.00) | 36.53(±0.00) | 19.20(±0.00) |
| | Wanda | 21.58(±0.50) | 32.51(±0.42) | 20.77(±0.18) | 33.38(±0.71) | 68.51(±0.34) | 51.60(±0.62) | 21.71(±0.39) |
| | SparseGPT | 17.45(±0.18) | 23.50(±0.70) | 17.27(±0.22) | 46.44(±0.62) | 69.61(±0.51) | 54.47(±0.33) | 23.14(±0.32) |
| | DSnoT | 23.69(±0.29) | 37.97(±0.99) | 23.04(±0.25) | 36.54(±0.35) | 67.57(±0.44) | 49.60(±0.54) | 21.60(±0.38) |
| | *ALPS* | **16.84**(±0.29) | **21.17**(±0.16) | **16.47**(±0.28) | **49.59**(±1.14) | **70.99**(±0.24) | **55.57**(±0.42) | **23.40**(±0.64) |
| OPT-6.7B | MP | 264.1(±0.0) | 205.8(±0.0) | 203.5(±0.0) | 1.46(±0.00) | 63.06(±0.00) | 42.51(±0.00) | 20.48(±0.00) |
| | Wanda | 16.08(±0.14) | 24.88(±0.25) | 17.04(±0.12) | 47.53(±0.89) | 71.34(±0.18) | 56.80(±0.43) | 24.68(±0.41) |
| | SparseGPT | 14.25(±0.16) | 18.39(±0.24) | 14.61(±0.22) | 58.90(±0.88) | 72.55(±0.34) | 59.99(±0.32) | 25.63(±0.42) |
| | DSnoT | 16.26(±0.16) | 23.32(±0.34) | 16.47(±0.09) | 51.41(±0.67) | 70.87(±0.16) | 56.00(±0.17) | 24.88(±0.42) |
| | *ALPS* | **13.54**(±0.10) | **17.51**(±0.19) | **13.93**(±0.25) | **61.47**(±1.21) | **73.26**(±0.25) | **61.03**(±0.36) | **26.59**(±0.64) |
| OPT-13B | MP | 485.0(±0.0) | 379.1(±0.0) | 257.7(±0.0) | 0.47(±0.00) | 60.50(±0.00) | 36.99(±0.00) | 21.08(±0.00) |
| | Wanda | 15.69(±0.11) | 20.73(±0.19) | 15.01(±0.06) | 50.56(±0.63) | 71.68(±0.29) | 58.44(±0.36) | 27.32(±0.25) |
| | SparseGPT | 12.98(±0.10) | 15.88(±0.10) | 13.39(±0.17) | 62.03(±0.78) | 73.25(±0.41) | 61.89(±0.58) | 28.74(±0.99) |
| | DSnoT | 15.28(±0.09) | 19.34(±0.08) | 15.21(±0.04) | 54.54(±0.70) | 72.43(±0.37) | 57.48(±0.43) | 26.25(±0.37) |
| | *ALPS* | **12.25**(±0.08) | **15.25**(±0.08) | **12.83**(±0.19) | **62.58**(±1.14) | **73.80**(±0.21) | **63.56**(±0.46) | **29.78**(±0.74) |
| OPT-30B | MP | 1981(±0) | 2061(±0) | 1656(±0) | 0.84(±0.00) | 58.22(±0.00) | 40.61(±0.00) | 18.94(±0.00) |
| | Wanda | 13.23(±0.40) | 16.95(±0.28) | 14.67(±0.18) | 52.86(±1.58) | 74.87(±0.14) | 64.21(±0.40) | 29.23(±0.37) |
| | SparseGPT | 10.90(±0.07) | 14.02(±0.10) | 12.04(±0.14) | 66.74(±0.49) | 75.59(±0.28) | 66.75(±0.55) | 31.23(±0.32) |
| | DSnoT | 12.36(±0.10) | 15.73(±0.07) | 13.55(±0.08) | 59.77(±0.51) | 74.94(±0.20) | 64.19(±0.19) | 29.44(±0.50) |
| | *ALPS* | **10.64**(±0.09) | **13.75**(±0.28) | **11.69**(±0.15) | **67.30**(±0.64) | **75.93**(±0.09) | **66.82**(±0.54) | **31.45**(±0.37) |
| LLaMA2-7B | MP | 37.76(±0.00) | 14864(±0) | 59.76(±0.00) | - | 70.57(±0.00) | 56.14(±0.00) | 30.55(±0.00) |
| | Wanda | 12.10(±0.07) | 86.02(±1.47) | 14.05(±0.08) | - | 70.10(±0.18) | 57.15(±0.38) | 29.80(±0.30) |
| | SparseGPT | 10.94(±0.08) | **75.97**(±4.97) | 12.04(±0.28) | - | 71.40(±0.34) | 59.93(±0.72) | 29.88(±0.75) |
| | DSnoT | 12.68(±0.09) | 102.0(±5.1) | 14.85(±0.11) | - | 69.31(±0.59) | 55.90(±0.42) | 26.81(±0.27) |
| | *ALPS* | **9.95**(±0.17) | 88.49(±9.20) | **10.87**(±0.42) | - | **72.83**(±0.56) | **60.42**(±0.68) | **31.69**(±0.68) |
| LLaMA2-13B | MP | 8.89(±0.00) | 203.8(±0.0) | 10.80(±0.00) | - | 71.49(±0.00) | 57.66(±0.00) | 30.38(±0.00) |
| | Wanda | 9.02(±0.04) | 88.93(±1.02) | 11.12(±0.05) | - | 73.16(±0.27) | 63.67(±0.54) | 33.92(±0.37) |
| | SparseGPT | 8.76(±0.10) | 64.82(±3.16) | 10.06(±0.26) | - | 73.94(±0.60) | 65.10(±0.86) | 35.31(±0.57) |
| | DSnoT | 9.21(±0.05) | 87.54(±2.04) | 11.43(±0.07) | - | 72.30(±0.23) | 63.38(±0.68) | 33.63(±0.63) |
| | *ALPS* | **8.14**(±0.10) | **52.34**(±2.08) | **9.36**(±0.33) | - | **74.72**(±0.58) | **65.71**(±0.59) | **35.96**(±0.50) |
| LLaMA3-8B | MP | 2403(±0.0) | 3434(±0.0) | 2372(±0.0) | - | 60.99(±0.0) | 40.03(±0.0) | 21.67(±0.0) |
| | Wanda | 24.36(±0.38) | 44.89(±1.12) | 30.81(±0.27) | - | 67.56(±0.32) | 56.20(±0.43) | 26.11(±0.70) |
| | SparseGPT | 16.35(±0.18) | 25.08(±0.61) | 18.89(±0.71) | - | 70.54(±0.70) | 63.09(±0.71) | 31.84(±0.59) |
| | DSnoT | 23.09(±0.42) | 40.95(±1.04) | 28.78(±0.14) | - | 67.70(±0.65) | 56.46(±0.38) | 25.68(±0.41) |
| | *ALPS* | **14.82**(±0.26) | **23.55**(±0.34) | **16.59**(±0.83) | - | **72.72**(±1.04) | **64.98**(±0.83) | **33.09**(±1.10) |

Table 16: Performance analysis for one-shot unstructured pruning of OPT models and LLaMA models at 2 : 4 sparsity. (↑): higher is better; (↓): lower is better. We omit LLaMA results on LAMBADA due to its poor performance without modifications.

| Model | Algorithm | WikiText2↓ | PTB↓ | C4↓ | LAMBADA↑ | PIQA↑ | ARC-Easy↑ | ARC-Challenge↑ |
|---|---|---|---|---|---|---|---|---|
| OPT-1.3B | MP | 240.1(±0.0) | 241.4(±0.0) | 90.84(±0.00) | 7.31(±0.00) | 61.97(±0.00) | 41.37(±0.00) | 19.62(±0.00) |
| | Wanda | 22.40(±0.22) | 32.41(±0.45) | 21.34(±0.19) | 35.57(±0.63) | 67.83(±0.40) | 50.78(±0.37) | 22.41(±0.45) |
| | SparseGPT | 20.23(±0.35) | 25.28(±0.20) | 19.42(±0.38) | 41.54(±0.97) | 65.85(±0.57) | 48.01(±0.74) | 21.95(±0.44) |
| | DSnoT | 23.69(±0.25) | 35.09(±0.34) | 22.74(±0.16) | 35.58(±0.99) | 67.06(±0.15) | 49.52(±0.34) | 21.76(±0.56) |
| | *ALPS* | **19.48**(±0.37) | **23.65**(±0.13) | **18.19**(±0.24) | **43.91**(±1.19) | **68.42**(±0.86) | **52.07**(±0.72) | **22.95**(±0.60) |
| OPT-2.7B | MP | 166.9(±0.0) | 195.3(±0.0) | 88.42(±0.00) | 10.74(±0.00) | 66.05(±0.00) | 45.29(±0.00) | 20.73(±0.00) |
| | Wanda | 17.04(±0.45) | 25.87(±0.20) | 17.14(±0.11) | 41.06(±0.53) | 70.79(±0.18) | 53.23(±0.19) | 23.26(±0.41) |
| | SparseGPT | 15.14(±0.24) | 20.36(±0.41) | 15.59(±0.15) | 49.32(±0.35) | 71.09(±0.20) | 54.55(±0.51) | 24.39(±0.38) |
| | DSnoT | 17.10(±0.24) | 24.72(±0.64) | 17.51(±0.09) | 44.60(±0.72) | 69.75(±0.16) | 52.61(±0.25) | 22.59(±0.45) |
| | *ALPS* | **14.75**(±0.16) | **18.90**(±0.15) | **15.18**(±0.18) | **52.66**(±1.08) | **71.46**(±0.19) | **56.96**(±0.50) | **25.03**(±0.34) |
| OPT-6.7B | MP | 196.2(±0.0) | 163.7(±0.0) | 160.4(±0.0) | 2.96(±0.00) | 65.61(±0.00) | 43.48(±0.00) | 20.90(±0.00) |
| | Wanda | 13.64(±0.12) | 20.93(±0.14) | 14.40(±0.06) | 57.75(±1.02) | 72.98(±0.39) | 60.08(±0.25) | 26.50(±0.43) |
| | SparseGPT | 12.63(±0.08) | 16.65(±0.32) | 13.42(±0.15) | 61.96(±1.13) | 73.22(±0.28) | 61.68(±0.44) | 26.45(±0.41) |
| | DSnoT | 13.59(±0.11) | 19.01(±0.27) | 14.43(±0.06) | 57.93(±0.70) | 72.88(±0.13) | 58.87(±0.33) | 26.35(±0.62) |
| | *ALPS* | **12.31**(±0.07) | **16.18**(±0.24) | **13.07**(±0.16) | **64.10**(±0.82) | **73.80**(±0.41) | **62.41**(±0.38) | **27.87**(±0.47) |
| OPT-13B | MP | 449.6(±0.0) | 366.9(±0.0) | 211.7(±0.0) | 1.54(±0.00) | 62.73(±0.00) | 41.46(±0.00) | 21.84(±0.00) |
| | Wanda | 13.37(±0.10) | 18.56(±0.20) | 13.57(±0.04) | 57.09(±0.77) | 73.80(±0.23) | 60.29(±0.18) | 27.94(±0.29) |
| | SparseGPT | 11.77(±0.09) | 14.76(±0.11) | 12.54(±0.10) | **64.41**(±0.96) | 73.66(±0.37) | 63.71(±0.42) | 29.56(±0.39) |
| | DSnoT | 13.07(±0.10) | 16.41(±0.10) | 13.57(±0.04) | 60.80(±0.81) | 72.86(±0.34) | 59.71(±0.24) | 28.24(±0.27) |
| | *ALPS* | **11.37**(±0.04) | **14.37**(±0.09) | **12.22**(±0.12) | 64.32(±0.83) | **74.31**(±0.32) | **64.65**(±0.39) | **30.51**(±0.28) |
| OPT-30B | MP | 564.1(±0.0) | 1487(±0) | 1005(±0) | 3.90(±0.00) | 62.84(±0.00) | 42.47(±0.00) | 22.27(±0.00) |
| | Wanda | 10.78(±0.08) | 14.07(±0.07) | 12.13(±0.03) | 61.94(±0.59) | 75.65(±0.18) | 66.90(±0.46) | 30.48(±0.41) |
| | SparseGPT | 10.30(±0.06) | 13.35(±0.15) | 11.52(±0.09) | **68.40**(±0.46) | 76.10(±0.25) | 67.88(±0.48) | **32.17**(±0.88) |
| | DSnoT | 10.83(±0.08) | 13.93(±0.04) | 12.16(±0.03) | 64.58(±0.55) | 75.42(±0.18) | 66.31(±0.44) | 30.58(±0.46) |
| | *ALPS* | **10.14**(±0.05) | **13.18**(±0.12) | **11.32**(±0.09) | 67.88(±0.49) | **76.61**(±0.50) | **67.89**(±0.25) | 32.01(±0.49) |
| LLaMA2-7B | MP | 15.91(±0.00) | - | 26.84(±0.00) | - | 71.93(±0.00) | 59.22(±0.00) | 32.85(±0.00) |
| | Wanda | 8.63(±0.01) | 46.90(±0.67) | 10.34(±0.02) | - | 73.74(±0.17) | 62.01(±0.18) | 31.47(±0.33) |
| | SparseGPT | 8.46(±0.03) | **45.03**(±1.13) | 9.75(±0.13) | - | 74.44(±0.26) | **63.80**(±1.25) | **33.52**(±0.66) |
| | DSnoT | 8.88(±0.01) | 48.00(±1.02) | 10.68(±0.01) | - | 73.34(±0.23) | 61.43(±0.44) | 30.96(±0.34) |
| | *ALPS* | **8.12**(±0.08) | 51.21(±2.78) | **9.29**(±0.20) | - | **74.78**(±0.17) | 63.56(±0.54) | 33.29(±0.56) |
| LLaMA2-13B | MP | 7.32(±0.00) | 137.3(±0.0) | 9.14(±0.00) | - | 74.43(±0.00) | 63.01(±0.00) | 35.67(±0.00) |
| | Wanda | 7.02(±0.01) | 55.07(±0.79) | 8.84(±0.02) | - | 75.69(±0.31) | 67.05(±0.13) | 38.26(±0.56) |
| | SparseGPT | 7.01(±0.03) | 47.64(±0.81) | 8.52(±0.14) | - | 75.84(±0.26) | 66.39(±0.84) | 38.74(±0.95) |
| | DSnoT | 7.13(±0.02) | 54.07(±0.88) | 8.95(±0.02) | - | 75.64(±0.23) | 67.33(±0.42) | 37.58(±0.63) |
| | *ALPS* | **6.81**(±0.07) | **42.15**(±0.89) | **8.22**(±0.17) | - | **76.52**(±0.57) | **69.41**(±0.69) | **39.22**(±0.56) |
| LLaMA3-8B | MP | 181.9(±0.0) | 347.9(±0.0) | 174.7(±0.0) | - | 65.18(±0.0) | 53.41(±0.0) | 26.37(±0.0) |
| | Wanda | 14.52(±0.08) | 24.26(±0.10) | 18.88(±0.11) | - | 71.52(±0.36) | 64.91(±0.39) | 34.03(±0.42) |
| | SparseGPT | 12.40(±0.13) | 17.90(±0.19) | 14.94(±0.38) | - | 73.20(±0.74) | 68.54(±0.82) | 34.86(±0.76) |
| | DSnoT | 14.76(±0.09) | 23.90(±0.20) | 18.89(±0.04) | - | 71.49(±0.23) | 65.65(±0.23) | 33.57(±0.81) |
| | *ALPS* | **11.57**(±0.12) | **16.57**(±0.38) | **13.76**(±0.47) | - | **74.57**(±0.40) | **69.65**(±0.61) | **37.20**(±0.79) |

Table 17: Performance analysis for one-shot unstructured pruning of OPT models and LLaMA models at 4 : 8 sparsity. (↑): higher is better; (↓): lower is better. We omit LLaMA results on LAMBADA due to its poor performance without modifications.

