# OpenReview forum: "ALPS: Improved Optimization for Highly Sparse One-Shot Pruning for Large Language Models"
_NeurIPS.cc/2024/Conference — NeurIPS 2024 poster_

### Official Review · Reviewer_1PwE · 2024-06-30

**Soundness:** 3
**Presentation:** 2
**Contribution:** 2
**Rating:** 3
**Confidence:** 3

**Summary:**

The paper introduces ALPS, an optimization-based framework for one-shot pruning of large language models (LLMs). ALPS leverages an ADMM-based algorithm with operator splitting and preconditioned conjugate gradient methods to achieve improvements in sparsity and perplexity over state-of-the-art methods, particularly in high-sparsity regimes.

**Strengths:**

**S1.**
ALPS achieves substantial reductions in test perplexity and improved zero-shot benchmark performance for highly sparse models.

**S2.**
Provides theoretical convergence guarantees for $\ell_0$-constrained optimization problems with ADMM solver.

**S3.**
Implements efficient post-processing techniques via conjugate projected gradient, enhancing computational performance.

**Weaknesses:**

**W1.**
Important Reference Missing: The paper does not cite "Fast and optimal weight update for pruned large language models" by Boža, which addresses a similar problem using an ADMM-based optimization algorithm. This omission is significant as both papers share highly similar problem definitions and solutions.

**W2.**
Limited Novelty: ALPS closely resembles methods discussed in both "Fast and optimal pruning" and "Progressive weight pruning of deep neural networks using ADMM." The primary difference is ALPS's specific application to LLMs. However, this differentiation might not be substantial enough to establish ALPS as a novel contribution. Btw, the reference of "Progressive weight pruning of deep neural networks using ADMM." seems also missing in the paper.

**W3.**
Performance at High Sparsity: At very high sparsity levels, ALPS’s perplexity remains significantly higher than the dense model. This indicates that the pruned LLMs by ALPS may still perform poorly and being practically useless, although they are better than those pruned by comparison methods.

**W4.**
Unfair Comparison: The comparison with methods like Wanda and DSnoT, which do not involve retraining after pruning, is solely based on perplexity. This is unfair because it overlooks the overall running time per iteration/epoch. ALPS’s performance should be compared with these methods by considering both perplexity and computational efficiency to provide a more balanced evaluation. In addition, I believe if Wanda or DSnoT is combined with some re-training techniques, they can achieve a much lower perplexity as well.

Reference:

Boža, Vladimír. "Fast and optimal weight update for pruned large language models." arXiv preprint arXiv:2401.02938 (2024).

Ye, Shaokai, et al. "Progressive weight pruning of deep neural networks using ADMM." arXiv preprint arXiv:1810.07378 (2018).

**Questions:**

Please refer to my points listed in the weakness section. Besides those,

1. Could you provide a clearer explanation of the practical implications of your theoretical contributions?

2. I encourage the authors to report the overall running time per epoch for ALPS in comparison with Wanda and DSnoT.

3. Could you provide the perplexity of the dense models (without any pruning) in all the tables in the paper?

4. How does ALPS's speed of convergence compare with the referenced methods, and what practical benefits does the novel penalty parameter update scheme offer?

**Limitations:**

The authors have not thoroughly discussed the limitations and potential negative societal impacts. They should address the limited novelty compared to existing ADMM-based methods and discuss the risks and mitigation strategies for the misuse of more accessible powerful models.

---

> ### Author Rebuttal · Authors · 2024-08-07
>
> **Reply to W1 and W2:** Thanks for these references—we will include them in our revised paper. ALPS differs significantly from the mentioned works as follows:
>
> + Comparison with "Progressive weight pruning of deep neural networks using ADMM":
> This paper applies ADMM to the original loss function, requiring expensive full network training via SGD. In contrast, ALPS performs post-training pruning, applying ADMM to layerwise reconstruction problems (least squares objective) with cardinality constraints. This approach enables us to scale ALPS to 30B parameters on a single V100 GPU.
>
> + Comparison with "Fast and optimal pruning" (Boža):  In this interesting work, Boža performs pruning post-training using a layerwise reconstruction loss. They apply ADMM to solve a least squares problem for a *given* sparsity mask.  Boža selects the sparsity mask via iterative magnitude pruning. Using ADMM speeds up the back-solve procedure.
>
>      + ALPS, in contrast, is an end-to-end approach that directly targets the $\ell_0$ constrained least squares problem with theoretical guarantees. **ALPS simultaneously optimizes over the weights and the sparsity pattern, unlike the work by Boža where ADMM is used to perform the backsolve operation.**  Therefore, ALPS achieves lower layerwise reconstruction loss than Boža, as shown in Table 1 in the general rebuttal.
>
>      + Since Boža employs ADMM for backsolving [finding the weights of the least squares problem given a mask], we also compare it with our PCG procedure, which serves as a backsolve method. We tested both approaches for computing weights on a given support. Results show that PCG outperforms Boža’s ADMM procedure in both computational time and objective value (see Table 2 in general rebuttal). The time advantage of PCG stems from its ability to backsolve without explicitly computing matrix inverses.
>
> + Novelty of ALPS:
> Please note that ALPS has several differences/innovations compared to standard ADMM:
>      - A novel penalty parameter updating scheme that enables finding high-quality support and accelerates convergence. As we show in our response to Q1 and Q4, in practice, our proposed ADMM has better convergence properties compared to the usual version of ADMM for $\ell_0$-constrained least squares problems.
>     - As far as we know, our theory is the **first** theoretical convergence result for ADMM applied to $\ell_0$-constrained problems.
>     - A PCG-based post-processing technique optimized with vectorization and GPU parallelism, achieving up to 200x speedup compared to direct backsolve for updating weights on a given support.
> These contributions, taken together, make our work significantly different from existing ADMM methods especially in the area of LLM pruning.
>
> **Reply to W3:** Please note that our study demonstrates ALPS's effectiveness as a pruning method across all sparsity levels, including 60% sparsity and N:M sparsity pattern. In these cases, the pruned models maintain competitive performance.
>
> You are right that model utility can deteriorate with one-shot pruning at very high sparsity levels — in such cases, we can recover model performance via retraining. Since ALPS-pruned models outperform models pruned by competing methods, they would require fewer retraining epochs to recover lost performance [1]. Consequently, ALPS remains valuable at high sparsity levels by reducing computational costs during the expensive retraining phase.
>
> [1] Meng, X., Ibrahim, S., Behdin, K., Hazimeh, H., Ponomareva, N., & Mazumder, R. OSSCAR: One-Shot Structured Pruning in Vision and Language Models with Combinatorial Optimization.
>
> **Reply to W4:** Thanks for your thoughtful comment. In our response to reviewer ZgXD, we report the running time of ALPS and other one-shot pruning methods. We emphasize that with retraining, the total runtimes of Wanda and DSnoT become much higher than ALPS. As reported in Wanda, fine-tuning LLaMa-7B with LoRA takes about 24 hours on a V100 GPU, and full parameter fine-tuning takes 24 days. In contrast, ALPS prunes the model in less than 1 hour. As a layer-wise pruning method, ALPS loads weights and activations onto the GPU layer by layer, enabling it to prune models as large as OPT-30B on a single V100 GPU, while retraining often requires more careful memory management. Given these considerations, we think that retraining methods would be much more computationally expensive than ALPS. Please recall that our focus here is on demonstrating the effectiveness of ALPS as a one-shot pruning method.
> As a side note, as we explained in our reply to ZgXD, the runtime of ALPS can be further decreased by loosening the convergence criteria of ALPS.
>
> **Reply to Q1 and Q4:** Our proposed $\rho$ update scheme, theoretically supported by Theorem 1, ensures that ALPS converges rapidly while finding high-quality solutions. In contrast, ADMM with a fixed $\rho$ may fail to converge when applied to $\ell_0$ constrained least squares problems. To demonstrate this, we compared ALPS with ADMM using fixed $\rho$ values. We examined two key metrics: reconstruction loss (objective) and the rate of change of the support (of weights) between consecutive iterations (this measures the convergence speed of the algorithm). The results are presented in the general rebuttal (Tabl 3 and 4). Results reveal that ADMM with a large $\rho(=3)$ converges quickly but yields poor solutions, while a small $\rho(=0.3)$ fails to converge. ALPS, utilizing our $\rho$ update scheme, achieves both rapid convergence and high-quality solutions.
>
> **Reply to Q2:** Thanks for your comment — please refer to our response to reviewer ZgXD.
>
> **Reply to Q3:** Please refer to Table 5 in general rebuttal. In the table, we omit LLaMA results on LAMBADA due to its poor performance without modifications.
>
> ---
>
> In light of the above discussions and results, we kindly ask the reviewer to consider increasing their evaluation.

---

> > ### Comment · Reviewer_1PwE · 2024-08-10
> >
> > I sincerely appreciate the authors' response, which addressed some of my previous concerns. However, given the current dispersion of ratings, I prefer to proceed cautiously and will consider the rebuttal more thoroughly in the next phase of discussion. I am inclined to maintain my current score for now but will remain open to adjusting it during the next phase.

---

> > > ### Author Response · Authors · 2024-08-14
> > >
> > > Thanks for your feedback and thoughtful comments. We appreciate your willingness and consideration to be open to revising your score. Please let us know if you have any additional questions or need clarification.

---

### Official Review · Reviewer_xarH · 2024-07-12

**Soundness:** 3
**Presentation:** 3
**Contribution:** 2
**Rating:** 7
**Confidence:** 3

**Summary:**

This work presents an LLM pruning framework that formulates the problem as finding a sparse weight matrix to reconstruct the layer-wise activations. This work incorporates the operator splitting technique and preconditioned conjugate gradient methods to solve the pruning problem. Experiments demonstrate that the proposed method achieves better single-layer reconstruction error and improved performance on downstream tasks.

**Strengths:**

This work is motivated by a clear theoretical rationale.

**Weaknesses:**

Experiments could be more solid since many experiments are run on OPT, which is somewhat out-of-date. It would also be better to consider more challenging benchmarks, such as GSM8K or other questions that require generating a long answer.

**Questions:**

It would be better to compare with other representative pruning methods beyond the one-shot unstructured pruning, and to explore how this method could be integrated with others.

This work discusses engineering optimizations, such as incorporating PCG with GPU. Will you open-source the code or provide more efficient results from these engineering efforts?

**Limitations:**

There is no section for limitation, but the author discusses some in the conclusion. One limitation is the setting of layer-wise activation reconstruction. Recent works, especially the activation-aware pruning methods, reported that activations are not equally important.

---

> ### Author Rebuttal · Authors · 2024-08-07
>
> We would like to thank the reviewer for their thoughtful feedback. Below, we provide some answers/clarifications.
>
> **Experiments could be more solid since many experiments are run on OPT, which is somewhat out-of-date. It would also be better to consider more challenging benchmarks, such as GSM8K or other questions that require generating a long answer.**
>
> **Reply:** We appreciate the reviewer's valuable suggestion. Based on your comment and that of reviewer ZgXD, we will consider comparing ALPS with other methods on the recent model LLaMA-3 and broaden our assessment by incorporating knowledge-intensive and reasoning-based tasks, specifically MMLU and GSM8K.
>
> **It would be better to compare with other representative pruning methods beyond the one-shot unstructured pruning, and to explore how this method could be integrated with others.**
>
> **Reply:** Thank you for this insightful suggestion! Beyond unstructured pruning, numerous representative pruning methods focus on structured sparsity patterns, such as block sparsity [1] and row sparsity [2]. Though direct numerical comparisons with these methods are potentially challenging due to their different settings, the idea of integrating ALPS with structured pruning methods is intriguing. Having a pruned model with *both* unstructured and structured sparsity could optimize both storage costs and inference time. It would be certainly interesting to explore such an integration as future work.
>
> [1] Gray, S., Radford, A., & Kingma, D. P. GPU kernels for block-sparse weights.
>
> [2] Meng, X., Ibrahim, S., Behdin, K., Hazimeh, H., Ponomareva, N., & Mazumder, R. OSSCAR: One-Shot Structured Pruning in Vision and Language Models with Combinatorial Optimization.
>
>
> **This work discusses engineering optimizations, such as incorporating PCG with GPU. Will you open-source the code or provide more efficient results from these engineering efforts?**
>
> **Reply:** If the paper is accepted, we will release the codes. Additionally, we will provide explanations of such implementations and a tutorial on their usage.

---

### Official Review · Reviewer_ZgXD · 2024-07-13

**Soundness:** 3
**Presentation:** 3
**Contribution:** 3
**Rating:** 6
**Confidence:** 3

**Summary:**

This paper introduces ALPS, a novel optimization-based framework for one-shot unstructured pruning of LLMs. The key contributions are:

- Formulating LLM pruning as an l0-constrained optimization problem solved using operator splitting (ADMM).
- A penalty parameter update scheme to accelerate convergence.
- A post-processing step using preconditioned conjugate gradient (PCG) to refine weights.
- Theoretical convergence guarantees for the proposed algorithm.

The authors demonstrate that ALPS outperforms state-of-the-art pruning methods, especially at high sparsity levels, on various LLMs including OPT and LLaMA models.

**Strengths:**

1. The proposed method directly addresses the pruning problem without relying on heuristics.
2. The ADMM algorithm introduced in the paper comes with theoretical convergence guarantees, adding reliability to the approach.
3. The paper includes comprehensive experiments on large-scale models (up to 30B parameters), demonstrating consistent improvements over existing methods.

**Weaknesses:**

1. It would be beneficial to include evaluations on more recent models such as LLaMA-3, instruction-tuned models, and extremely large models (70B+ parameters) to demonstrate the method's applicability to the latest advancements in the field.
2. The current evaluation tasks are relatively limited. Expanding the evaluation to include knowledge-intensive tasks (e.g., MMLU) and reasoning-based tasks (e.g., GSM8K and HumanEval) would provide a more comprehensive assessment of the method's effectiveness.
3. The current focus is primarily on N:M sparsity patterns. It would be great to explore structured pruning, which could lead to more significant improvements in inference speed, making the approach more versatile and practical for real-world applications.

**Questions:**

1. How does the runtime of ALPS compare to other methods, especially for very large models?
2. Can (how can) ALPS be extended to handle other types (structured sparsity) of structured sparsity patterns beyond N:M?

**Limitations:**

The authors discussed some limitations of their work e.g. extending ALPS to incorporate structured pruning constraints and quantization.

---

> ### Author Rebuttal · Authors · 2024-08-07
>
> We would like to thank the reviewer for their thoughtful feedback. Below, we provide some answers/clarifications.
>
> **It would be beneficial to include evaluations on more recent and extremely large models to demonstrate the method's applicability. The current evaluation tasks are relatively limited. Expanding the evaluation to include knowledge-intensive tasks and reasoning-based tasks would provide a more comprehensive assessment.**
>
> **Reply:** We appreciate the reviewer's valuable suggestion. We will consider comparing ALPS with other methods on the recent model LLaMA-3 and broaden our assessment by incorporating knowledge-intensive and reasoning-based tasks, specifically MMLU and GSM8K. As a side remark, given our academic resource constraints (we conduct our experiments on a V100 GPU with 32GB memory), the largest model we’ve tried in the paper is OPT-30B.
>
> **The current focus is primarily on N:M sparsity patterns. It would be great to explore structured pruning. Can ALPS be extended to handle other types (structured sparsity) of structured sparsity patterns beyond N:M?**
>
> **Reply:**  While we focus on unstructured sparsity and N:M sparsity, our approach can be generalized to handle other structured sparsity patterns, as discussed below.
>
> Let $S$ denote the set of weight matrices with the desired structured sparsity pattern (in the case of unstructured sparsity, $S$ contains all matrices with no more than $k$ non-zero elements). Our method can be extended to find the optimal matrix in $S$ by modifying the $\mathbf{D}$-update step in Eq.(4) in our paper. We can set $\mathbf{D}^{(t+1)}$ to be the projection of $\mathbf{W}^{(t+1)}+\mathbf{V}^{(t+1)}/\rho$ onto $S$. Our approach remains efficient and can find high-quality solutions for sparsity patterns with low-cost projection operations, including:
> + Block sparsity [1]
> + Hierarchical sparsity [2]
> + Row sparsity [3]
>
> Importantly, our convergence results (Theorem 1) and proposed PCG step directly apply to these structured sparsity patterns as well.
>
> We thank the reviewer for bringing up this point, and we will be happy to discuss these extensions in our revision.
>
> [1] Gray, S., Radford, A., & Kingma, D. P. GPU kernels for block-sparse weights.
>
> [2] Wu, Y. N., Tsai, P. A., Muralidharan, S., Parashar, A., Sze, V., & Emer, J. Highlight: Efficient and flexible DNN acceleration with hierarchical structured sparsity.
>
> [3] Meng, X., Ibrahim, S., Behdin, K., Hazimeh, H., Ponomareva, N., & Mazumder, R. OSSCAR: One-Shot Structured Pruning in Vision and Language Models with Combinatorial Optimization.
>
>
> **How does the runtime of ALPS compare to other methods, especially for very large models?**
>
> **Reply:**  We compare the runtime of ALPS with other methods in the following table. Here, the runtime includes the time for generating input activations $\mathbf{X}$.
>
> |Method| OPT-1.3B | OPT-2.7B | OPT-6.7B | OPT-13B | OPT-30B | LLaMA-7B|LLaMA-13B|
> |-|-|-|-|-|-|-|-|
> | MP       |4.7|8.9|23|47|120|25|46|
> | Wanda    |99|161|280|502|1027|214|407|
> | DSnoT    |125|213|417|758|1528|347|651|
> | SparseGPT|363|728|1621|2980|6662|1263|2319|
> | ALPS     |963|2360|6069|14323|48366|3043|7145|
> | ALPS-simple|297|599|1375|2470|5021|1013|2595|
>
>
> |Method |MP | Wanda | DSnoT |SparseGPT | ALPS  | ALPS-simple|
> |-|-|-|-|-|-|-|
> |**Loss**|1.98e-1|2.26e-1|2.06e-1|8.54e-2|5.32e-2 |7.52e-2 |
>
> ALPS employs an advanced optimization method to solve the layerwise reconstruction problem, which results in longer running times compared to other algorithms. However, it's important to note that ALPS's runtime is still negligible when compared to fine-tuning methods for LLMs. For context, Wanda [1] reports that fine-tuning LLaMa-7B with LoRA takes about 24 hours on a V100 GPU, while full parameter fine-tuning takes 24 days. In contrast, ALPS prunes the model in less than 1 hour.
> Furthermore, in our paper, ALPS is run with a tight convergence criterion. We can improve its runtime further by using a variant, ALPS-simple, which uses a looser convergence criterion. As shown in the above tables, ALPS-simple achieves shorter running times compared to SparseGPT while still maintaining better utility, as measured by single-layer reconstruction loss. We are happy to include the performance results of ALPS-simple in the revised paper if it is of interest to the reviewers.
>
> [1] Sun, M., Liu, Z., Bair, A., & Kolter, J. Z. A simple and effective pruning approach for large language models.

---

> > ### Comment · Reviewer_ZgXD · 2024-08-12
> > **Response to rebuttals**
> >
> > Thank you for your detailed responses. The paper is in good shape, but still have some concerns remain unsolved:
> >
> > 1. I understand the resource constraints, but some results on models like LLaMA-3 or a discussion on scaling to 70B+ parameters would be valuable. OPT-30B is relatively old, and to my knowledge, it's sometimes easier to prune them than more recent 'overtrained' models, e.g. Llama-3 and Qwen 1.5 / 2.
> >
> > 2. Including knowledge-intensive (e.g., MMLU) and reasoning-based tasks (e.g., GSM8K) would strengthen the paper.
> >
> > 3. Theoretical analysis of extension is promising, and empirical results or case studies on these structured sparsity patterns would provide practical insights.

---

> > > ### Author Response · Authors · 2024-08-14
> > >
> > > Thank you for your positive feedback. Based on your valuable comment, we have conducted further experiments to compare ALPS with SparseGPT, currently our strongest competitor, on LLaMA3-7B at 50%, 70%, and 2:4 sparsity levels. The results, as shown in the table below, demonstrate that ALPS outperforms SparseGPT on recent models LLaMA3.
> > >
> > > In response to your suggestions, we are currently running extended experiments to compare ALPS on pruning LLaMA3 with other competitors [in the current paper] across a range of sparsity levels. We will include our current results and new results in a revision.
> > >
> > >
> > > | 2:4 |Lambada|Piqa|Arc-easy|Arc-challenge|
> > > |---|---|---|---|---|
> > > |SparseGPT|36.93|70.40|59.01|28.84|
> > > |ALPS|42.79|70.89|61.83|28.92|
> > >
> > > | 50% |Lambada|Piqa|Arc-easy|Arc-challenge|
> > > |---|---|---|---|---|
> > > |SparseGPT|61.87|75.57 |72.39|39.25|
> > > |ALPS|64.04|76.93|73.40|40.61|
> > >
> > > | 70% |Lambada|Piqa|Arc-easy|Arc-challenge|
> > > |---|---|---|---|---|
> > > |SparseGPT|8.13|61.21|41.54|20.05|
> > > |ALPS|16.42|64.58|46.89|21.84|
> > >
> > > Additionally, we plan to include evaluation results for MMLU and GSM8K to further enhance the paper. Thanks again for your thoughtful feedback, which has strengthened our work.

---

### Author Rebuttal · Authors · 2024-08-07

We thank all reviewers for their thoughtful comments. In response to their comments, we have conducted additional numerical comparisons and included additional discussions in relation to the existing work. We present a summary below:

+ Comparison with existing methods & Novelty [see Ref 1PwE]: We've clarified key differences between ALPS and prior works on ADMM. In particular, we explain how our approach differs from a direct application of ADMM, and results in better performance. We’ve provided supporting numerical evidence.

+ New experimental results. Based on referee feedback,  we perform additional numerical experiments in the rebuttal:

   - [see Ref 1PwE] As a backsolve method, our proposed PCG outperforms Boža’s ADMM method in both computational time and objective value. ALPS also achieves lower layerwise loss compared to Boža.
   - [see Ref 1PwE] Our proposed parameter updating scheme enables ALPS to converge rapidly while finding high-quality solutions. In contrast, standard ADMM may fail to converge.
   - [see Ref ZgXD, 1PwE] While ALPS is already quite efficient, we can further accelerate ALPS by using a loose convergence criterion. This results in shorter running times compared to SparseGPT while still maintaining better utility.

+ Potential extensions [see Ref ZgXD]: We discuss how ALPS can be generalized to handle other structured sparsity patterns, including block, hierarchical, and row sparsity.

 ---

We provide some tables here for space reasons. We refer to these tables in our response to the reviewer 1PwE.

**Table 1:** The layerwise reconstruction loss on a single layer.

| Sparsity  | 0.4|  0.5 | 0.6  | 0.7 | 0.8 | 0.9|
|-|-|-|-|-|-|-|
|ALPS|3.55e-3|7.56e-3|1.47e-2|2.77e-2|5.32e-2|1.13e-1|
|Boža|4.47e-3|9.53e-3|1.83e-2|3.36e-2|6.19e-2|1.25e-1|

**Table 2:** The layerwise reconstruction loss and runtime for applying PCG and Boža to find the optimal weights on a given support.

| Sparsity  | 0.4|  0.5 | 0.6  | 0.7 | 0.8 | 0.9|
|-|-|-|-|-|-|-|
|ALPS-PCG(Loss)|5.0e-3|1.12e-2|2.27e-2|4.39e-2|8.51e-2|1.78e-1|
|Boža(Loss)|5.1e-3|1.12e-2|2.32e-2|4.49e-2|8.83e-2|2.00.e-1|
|ALPS-PCG(Time)|0.01s|0.01s|0.01s|0.01s|0.01s|0.01s|
|Boža(Time)|0.11s|0.11s|0.11s|0.11s|0.11s|0.11s|

**Table 3:** The reconstruction loss (objective) over iterations, comparing ALPS with ADMM using a fixed penalty parameter $\rho$.

| Loss/Iter   |5 | 10  | 20  | 30| 50| 100|
|---|---|---|---|---|---|---|
| ALPS |1.63e-1 | 1.28e-1  | 5.95e-2  | 5.32e-2  |5.31e-2  |5.31e-2  |
| ADMM($\rho=0.3$)  | 7.83e-2  | 7.55e-2  | 7.50e-2 |  7.47e-2 | 7.47e-2 | 7.45e-2  |
| ADMM($\rho=3$)  |  9.32e-2 | 8.18e-2  |7.64e-2 |  7.53e-2 | 7.45e-2   |7.42e-2 |

**Table 4:** The rate of change of the support (of weights) between consecutive iterations, comparing ALPS with ADMM using a fixed penalty parameter $\rho$.

| Supp change / Iter   |5 | 10  | 20  | 30| 50 | 100|
|---|---|---|---|---|---|---|
| ALPS |20.2% | 17.0% | 2.8% | 0.0% | 0.0% | 0.0% |
| ADMM($\rho=0.3$)  | 6.4%  | 7.0%  | 7.0%  |7.0%  | 6.9%| 6.9%|
| ADMM($\rho=3$)  | 0.2%  |  <0.1% | <0.1%  |<0.1%   | <0.1% | <0.1% |

**Table 5:** Perplexity and zero-shot evaluation results for dense models.

| Model |  WikiText2↓| PTB ↓ | C4 ↓ | LAMBADA↑ |PIQA↑ |ARC-Easy↑| ARC-Challenge↑|
|---|---|---|---|---|---|---|---|
| OPT-1.3B  |14.63|20.29|16.07|58.80|72.36|50.93|29.44|
| OPT-2.7B  |12.47|17.97|14.34|64.82|74.81|54.34|31.31|
| OPT-6.7B  |10.86|15.77|12.71|68.72|76.39|60.14|34.56|
| OPT-13B   |10.13|14.52|12.06|70.23|76.88|61.83|35.75|
| OPT-30B   |9.56 |14.04|11.44|72.39|78.18|65.40|38.14|
| LLaMA-7B  |5.47|22.51|6.97|-|78.29|69.23|39.93|
| LLaMA-13B |4.88|28.87|6.47|-|78.78|73.27|45.56|

---

### Decision · Program_Chairs · 2024-09-25

**Decision:**

Accept (poster)

**Comment:**

This paper proposed a large language model (LLM) pruning framework, which turns the problem into  layer-wise activation reconstruction problem via sparse weight matrix. The operator splitting technique and preconditioned conjugate gradient methods are deployed and the empirical studies show the better single-layer reconstruction error and correspondingly the improved performance on downstream tasks.

Overall the reviewers like the theoretical convergence guarantees and the practical applicability of the proposed method. There has been some major concerns from reviewer 1PwE, but after the rebuttal some of them have been addressed. We highly encourage the authors to incorporate the additional artifacts into the revisions to make the paper more solid.